# NEURAL DESIGN FOR GENETIC PERTURBATION EXPERIMENTS

**Aldo Pacchiano**
Microsoft Research NYC
apacchiano@microsoft.com

**Drausin Wulsin & Robert A. Barton & Luis Voloch**
Immunai
{drausin,robert.barton,luis}@immunai.com

## ABSTRACT

The problem of how to genetically modify cells in order to maximize a certain cellular phenotype has taken center stage in drug development over the last few years (with, for example, genetically edited CAR-T, CAR-NK, and CAR-NKT cells entering cancer clinical trials). Exhausting the search space for all possible genetic edits (perturbations) or combinations thereof is infeasible due to cost and experimental limitations. This work provides a theoretically sound framework for iteratively exploring the space of perturbations in pooled batches in order to maximize a target phenotype under an experimental budget. Inspired by this application domain, we study the problem of batch query bandit optimization and introduce the Optimistic Arm Elimination (OAE) principle designed to find an almost optimal arm under different functional relationships between the queries (arms) and the outputs (rewards). We analyze the convergence properties of OAE by relating it to the Eluder dimension of the algorithm's function class and validate that OAE outperforms other strategies in finding optimal actions in experiments on simulated problems, public datasets well-studied in bandit contexts, and in genetic perturbation datasets when the regression model is a deep neural network. OAE also outperforms the benchmark algorithms in 3 of 4 datasets in the GeneDisco experimental planning challenge.

## 1 INTRODUCTION

We are inspired by the problem of finding the genetic perturbations that maximize a given function of a cell (a particular biological pathway or mechanism, for example the proliferation or exhaustion of particular immune cells) while performing the least number of perturbations required. In particular, we are interested in prioritizing the set of genetic knockouts (via shRNA or CRISPR) to perform on cells that would optimize a particular scalar cellular phenotype. Since the space of possible perturbations is very large (with roughly 20K human protein-coding genes) and each knockout is expensive, we would like to order the perturbations strategically so that we find one that optimizes the particular phenotype of interest in fewer total perturbations than, say, just brute-force applying all possible knockouts. In this work we consider only single-gene knockout perturbations since they are the most common, but multi-gene perturbations are also possible (though considerably more technically complex to perform at scale). While a multi-gene perturbation may be trivially represented as a distinct (combined) perturbation in our framework, we leave for future work the more interesting extension of embedding, predicting, and planning these multi-gene perturbations using previously observed single-gene perturbations.

With this objective in mind we propose a simple method for improving a cellular phenotype under a limited budget of genetic perturbation experiments. Although this work is inspired by this concrete biological problem, our results and algorithms are applicable in much more generality to the setting of experimental design with neural network models. We develop and evaluate a family of algorithms for the zero noise batch query bandit problem based on the Optimistic Arm Elimination principle (OAE). We focus on developing tractable versions of these algorithms compatible with neural network function approximation.

During each time-step OAE fits a reward model on the observed responses seen so far while at the same time maximizing the reward on all the arms yet to be pulled. The algorithm then queries the batch of arms whose predicted reward is maximal among the arms that have not been tried out.

We conduct a series of experiments on synthetic and public data from the UCI Dua & Graff (2017) database and show that OAE is able to find the optimal "arm" using fewer batch queries than other algorithms such as greedy and random sampling. Our experimental evaluation covers both neurally realizable and not neurally realizable function landscapes. The performance of OAE against benchmarks is comparable in both settings, demonstrating that although our presentation of the OAE algorithm assumes realizability for the sake of clarity, it is an assumption that is not required in practice. In the setting where the function class is realizable i.e. the function class $\mathcal{F}$ used by OAE contains the function generating the rewards, and the evaluation is noiseless we show two query lower bounds for the class of linear and $1-$Lipshitz functions.

We validate OAE on the public CMAP dataset Subramanian et al. (2017), which contains tens of thousands of genetic shRNA knockout perturbations, and show that it always outperforms a baseline and almost always outperforms a simpler greedy algorithm in both convergence speed to an optimal perturbation and the associated phenotypic rewards. These results illustrate how perturbational embeddings learned from one biological context can still be quite useful in a different biological context, even when the reward functions of these two contexts are different. Finally we also benchmark our methods in the GeneDisco dataset and algorithm suite (see Mehrjou et al. (2021)) and show OAE to be competitive against benchmark algorithms in the task of maximizing HitRatios.

## 2 RELATED WORK

**Bayesian Optimization** The field of Bayesian optimization has long studied the problem of optimizing functions severely limited by time or cost Jones et al. (1998). For example, Srinivas et al. (2009) introduce the GP-UCB algorithm for optimizing unknown functions. Other approaches based on adaptive basis function regression have also been used to model the payoff function as in Snoek et al. (2015). These algorithms have been used in the drug discovery context. Mueller et al. (2017) applied Bayesian optimization to the problem of optimizing biological phenotypes. Very recently, GeneDisco was released as a benchmark suite for evaluating active learning algorithms for experiment design in drug discovery Mehrjou et al. (2021). Perhaps the most relevant to our setting are the many works that study the batch acquisition setting in Bayesian active learning and optimization such as Kirsch et al. (2019); Kathuria et al. (2016) and the $\mathrm{GP-BUCB}$ algorithm of Desautels et al. (2014). In this work we move beyond the typical parametric and Bayesian assumptions from these works towards algorithms that work in conjunction with neural network models. We provide guarantees for the no noise setting we study based on the Eluder dimension Russo & Van Roy (2013).

**Parallel Bandits** Despite its wide applicability in many scientific applications, batch learning has been studied relatively seldom in the bandit literature. Despite this, recent work (Chan et al., 2021) show that in the setting of contextual linear bandits (Abbasi-Yadkori et al., 2011), the finite sample complexity of parallel learning matches that of sequential learning irrespective of the batch size provided the number of batches is large enough. Unfortunately, this is rarely the regime that matters in many practical applications such as drug development where the size of the experiment batch may be large but each experiment may be very time consuming, thus limiting their number. In this work we specifically address this setting in our experimental evaluation in Section E.

**Structure Learning** Prior work in experiment design tries to identify causal structures with a fixed budget of experiments Ghassami et al. (2018). Scherrer et al Scherrer et al. (2021) proposes a mechanism to select intervention targets to enable more efficient causal structure learning. Sussex et al. (2021) extend the amount of information contained in each experiment by simultaneously intervening on multiple variables. Causal matching, where an experimenter can perform a set of interventions aimed to transform the system to a desired state, is studied in Zhang et al. (2021).

**Neural Bandits** Methods such as Neural UCB and Shallow Neural UCB Zhou et al. (2020); Xu et al. (2020) are designed to add an optimistic bonus to model predictions of a nature that can be analytically computed as is extremely reminiscent of the one used in linear bandits (Auer, 2002; Dani et al., 2008), thus their theoretical validity depends on the 'linearizing' conditions to hold. More

recently (Pacchiano et al., 2021b) have proposed the use of pseudo-label optimisim for the Bank Loan problem where they propose an algorithm that adds optimism to neural network predictions through the addition of fake data and is only analyzed in the classification setting. Our algorithms instead add optimism to their predictions. The later is achieved via two methods, either by explicitly encouraging it to fit a model whose predictions are large in unseen data, or by computing uncertainties.

**Active Learning**    Active learning is relatively well studied problem Settles (2009); Dasgupta (2011); Hanneke et al. (2014) particularly in the context of supervised learning. See for example Balcan et al. (2009), Dasgupta et al. (2007), Settles (2009), Hanneke et al. (2014). There is a vast amount of research on active learning for classification (see for example Agarwal (2013), Dekel et al. (2010) and Cesa-Bianchi et al. (2009) where the objective is to learn a linearly parameterized response model $\mathbb{P}(y|x)$. Broadly speaking there are two main sample construction approaches, diversity Sener & Savarese (2017); Geifman & El-Yaniv (2017); Gissin & Shalev-Shwartz (2019) and uncertainty sampling Tong & Koller (2001); Schohn & Cohn (2000); Balcan et al. (2009); Settles et al. (2007), successful in the large Guo & Schuurmans (2007); Wang & Ye (2015); Chen & Krause (2013); Wei et al. (2015); Kirsch et al. (2019) and small batch sizes regimes respectively. Diversity sampling methods produce spread out samples to better cover the space while uncertainty-based methods estimate model uncertainty to select what points to label. Hybrid approaches are common as well. A common objective in the active learning literature is to collect enough samples to produce a model that minimizes the population loss over the data distribution. This is in contrast with the objective we study in this work, which is to find a point in the dataset with a large response. There is a rich literature dedicated to the development of active learning algorithms for deep learning applications both in the batch and single sample settings Settles et al. (2007); Ducoffe & Precioso (2018); Beluch et al. (2018); Ash et al. (2021).

## 3   PROBLEM DEFINITION

Let $y_\star : \mathcal{A} \to \mathbb{R}$ be a response function over $\mathcal{A} \subset \mathbb{R}^d$. We assume access to a function class $\mathcal{F} \subset \mathrm{Fun}(\mathcal{A}, \mathbb{R})$ where $\mathrm{Fun}(\mathcal{A}, \mathbb{R})$ denotes the set of functions from $\mathcal{A}$ to $\mathbb{R}$. Following the typical online learning terminology we call $\mathcal{A}$ the set of arms. In this work we allow $\mathcal{A}$ to be infinite, although we only consider finite $\mathcal{A}$ in practice.

In our setting the experiment designer (henceforth called the learner) interacts with $y_\star$ and $\mathcal{A}$ in a sequential manner. During the $t-$th round of this interaction, aided by $\mathcal{F}$ and historical query and response information the learner is required to query a batch of $b \in \mathbb{N}$ arms $\{a_{t,i}\}_{i=1}^b \subset \mathcal{A}$ and observe *noiseless* responses $\{y_{t,i} = y_\star(a_{t,i})\}_{i=1}^b$ after which these response values are added to the historical dataset $\mathcal{D}_{t+1} = \mathcal{D}_t \cup \{(a_{t,i}, y_{t,i})\}_{i=1}^b$.

In this work we do not assume that $y_\star \in \mathcal{F}$. Instead we allow the learner access to a function class $\mathcal{F}$ to aid her in producing informative queries. This is a common situation in the setting of neural experiment design, where we may want to use a DNN model to fit the historical responses and generate new query points without prior knowledge of whether it accurately captures $y_\star$. Our objective is to develop a procedure that can recover an 'almost optimal' arm $a \in \mathcal{A}$ in the least number of arm pulls possible. We consider the following objective,

$\tau-$**quantile optimality.** Find an arm $a_\tau \in \mathcal{A}$ belonging to the top $\tau-$quantile[1] of $\{y_\star(a)\}_{a \in \mathcal{A}}$.

Although $\epsilon-$optimality (find an arm $a_\epsilon \in \mathcal{A}$ such that $y_\star(a_\epsilon) + \epsilon \geq \max_{a \in \mathcal{A}} y_\star(a)$ for $\epsilon \geq 0$) is the most common criterion considered in the optimization literature, for it to be meaningful it requires knowledge of the scale of $\max_{a \in \mathcal{A}} y_\star(a)$. In some scenarios this may be hard to know in advance. Thus in our experiments we focus on the setting of $\tau-$quantile optimality as a more relevant practical performance measure. This type of objective has been considered by many works in the bandit literature (see for example Szorenyi et al. (2015); Zhang & Ong (2021)). Moreover, it is a measure of optimality better related to practical objectives used in experiment design evaluation, such as hit ratio in the GeneDisco benchmark library Mehrjou et al. (2021). We show in Section E that our algorithms are successful at producing almost optimal arms under this criterion after a small number of queries. The main challenge we are required to overcome in this problem is designing a smart choice of batch

---

[1]In the case of an infinite set $\mathcal{A}$ quantile optimality is defined with respect to a measure over $\mathcal{A}$.

queries $\{a_{t,i}\}_{i=1}^B$ that balances the competing objectives of exploring new regions of the arm space and zooming into others that have shown promising rewards.

In this work we focus on the case where the observed response values $y_\star(a)$ of any arm $a \in \mathcal{A}$ are noiseless. In the setting of neural perturbation experiments the responses are the average of many expression values across a population of cells, and thus it is safe to assume the observed response is almost noiseless. In contrast with the noisy setting, when the response is noiseless, querying the same arm twice is never necessary. We leave the question on how to design algorithms for noisy responses in the function approximation regime for future work. although note it can be reduced to our setting if we set the exploitation round per data point sufficiently large.

**Evaluation.** After the queries $\cup_{\ell=1}^t \{a_{\ell,i}\}_{i=1}^b$ the learner will output a candidate approximate optimal arm $\hat{a}_t$ among all the arms whose labels she has queried (all arms in $\mathcal{D}_{t+1}$) by considering $(\hat{a}_t, \hat{y}_t) = \arg\max_{(a,y) \in \mathcal{D}_{t+1}} y$, the point with the maximal observed reward so far. Given a quantile value $\tau$ we measure the performance of our algorithms by considering the first timestep $t_{\text{first}}^\tau$ where a $\tau-$quantile optimal point $\hat{a}_t$ was proposed.

## 4 OPTIMISTIC ARM ELIMINATION

With these objectives in mind, we introduce a family of algorithms based on the Optimistic Arm Elimination Algorithm (OAE) principle. We call $\mathcal{U}_t$ to the subset of arms yet to be queried by our algorithm. At time $t$ any OAE algorithm produces a batch of query points of size $b$ from[2] $\mathcal{U}_t$. Our algorithms start round $t$ by fitting an appropriate response predictor $\widetilde{f}_t : \mathcal{U}_t \to \mathbb{R}$ based on the historical query points $\mathcal{D}_t$ and their observed responses so far. Instead of only fitting the historical responses with a square loss and produce a prediction function $\widetilde{f}_t$, we encourage the predictions of $\widetilde{f}_t$ to be optimistic on the yet-to-be-queried points of $\mathcal{U}_t$.

We propose two *tractable* ways of achieving this. First by fitting a model (or an ensemble of models) $\widetilde{f}_t^o$ to the data in $\mathcal{D}_t$ and explicitly computing a measure of uncertainty $\tilde{u}_t : \mathcal{U}_t \to \mathbb{R}$ of its predictions on $\mathcal{U}_t$. We define the optimistic response predictor $\widetilde{f}_t(a) = \widetilde{f}_t^o(a) + \tilde{u}_t(a)$. Second, we achieve this by defining $\widetilde{f}_t$ to be the approximate solution of a constrained objective,

$$\widetilde{f}_t \in \arg\max_{f \in \mathcal{F}} \mathbb{A}(f, \mathcal{U}_t) \qquad \text{s.t.} \sum_{(a,y) \in \mathcal{D}_t} (f(a) - y)^2 \leq \gamma_t. \qquad (1)$$

where $\gamma_t$ a possibly time-dependent parameter satisfying $\gamma_t \geq 0$ and $\mathbb{A}(f, \mathcal{U})$ is an acquisition objective tailored to produce an informative arm (or batch of arms) from $\mathcal{U}_t$. We consider a couple of acquisition objectives $\mathbb{A}_{\text{avg}}(f, \mathcal{U}) = \frac{1}{|\mathcal{U}|} \sum_{a \in \mathcal{U}} f(a)$, $\mathbb{A}_{\text{hinge}}(f, \mathcal{U}) = \frac{1}{|\mathcal{U}|} \sum_{a \in \mathcal{U}} (\max(0, f(a)))^p$ for some $p > 0$ and $\mathbb{A}_{\text{softmax}}(f, \mathcal{U}) = \log\left(\sum_{a \in \mathcal{U}} \exp(f(a))\right)$ and $\mathbb{A}_{\text{sum}}(f, \mathcal{U}) = \sum_{a \in \mathcal{U}} f(a)$. An important acquisition functions of theoretical interest, although hard to optimize in practice are $\mathbb{A}_{\text{max}}(f, \mathcal{U}) = \max_{a \in \mathcal{U}} f(a)$ and its batch version $\mathbb{A}_{\text{max},b}(f, \mathcal{U}) = \max_{\mathcal{B} \subset \mathcal{U}, |\mathcal{B}| = b} \sum_{a \in \mathcal{B}} f(a)$. Regardless of whether $\widetilde{f}_t$ was computed via Equation 1 or it is an uncertainty aware objective of the form $\widetilde{f}_t(a) = \widetilde{f}_t^o(a) + \widetilde{u}_t(a)$, our algorithm then produces a query batch $B_t$ by solving

$$B_t \in \arg\max_{\mathcal{B} \subset \mathcal{U}_t, |\mathcal{B}| = b} \mathbb{A}_{\text{avg}}(\widetilde{f}_t, \mathcal{B}). \qquad (2)$$

The principle of Optimism in the Face of Uncertainty (OFU) allows OAE algorithms to efficiently explore new regions of the space by acting greedily with respect to a model that fits the rewards of the arms in $\mathcal{D}_t$ as accurately as possible but induces large responses from the arms she has not tried. If $y_\star \in \mathcal{F}$, and $\widehat{f}_t$ is computed by solving Equation 1, it can be shown the optimistic model overestimates the true response values i.e. $\sum_{a \in B_t} \widetilde{f}_t(a) \geq \sum_{a \in B_{\star,t}} y_\star(a)$ where $B_{\star,t} = \arg\max_{\mathcal{B} \subset \mathcal{U}_t, |\mathcal{B}| = b} \sum_{a \in \mathcal{B}} y_\star(a)$. Consult Appendix D.2 for a proof and an explanation of the relevance of this observation.

Acting greedily based on an optimistic model means the learner tries out the arms that may achieve the highest reward according to the current model plausibility set. After pulling these arms, the learner can successfully update the model plausibility set and repeat this procedure.

---

[2] The batch equals $\mathcal{U}_t$ when $|\mathcal{U}_t| \leq b$.

---

**Algorithm 1** Optimistic Arm Elimination Principle (OAE)

---

**Input** Action set $\mathcal{A} \subset \mathbb{R}^d$, num batches $N$, batch size $b$
**Initialize** Unpulled arms $\mathcal{U}_1 = \mathcal{A}$. Observed points and labels dataset $\mathcal{D}_1 = \emptyset$
**for** $t = 1, \cdots, N$ **do**
$\quad$ **If** $t = 1$ **then:**
$\quad$ · Sample uniformly a size $b$ batch $B_t \sim \mathcal{U}_1$.
$\quad$ **Else:**
$\quad$ · Solve for $\widetilde{f}_t$ and compute $B_t \in \arg\max_{\mathcal{B} \subset \mathcal{U}_t || \mathcal{B} | = b} \mathbb{A}(\widetilde{f}_t, \mathcal{B})$.
$\quad$ Observe batch rewards $Y_t = \{y_*(a) \text{ for } a \in B_t\}$
$\quad$ Update $\mathcal{D}_{t+1} = \mathcal{D}_t \cup \{(B_t, Y_t)\}$ and $\mathcal{U}_{t+1} = \mathcal{U}_t \backslash B_t$ .

---

## 4.1 TRACTABLE IMPLEMENTATIONS OF OAE

In this section we go over the algorithmic details behind the approximations that we have used when implementing the different OAE methods we have introduced in Section 4.

### 4.1.1 OPTIMISTIC REGULARIZATION

In order to produce a tractable implementation of the constrained problem 1 we approximate it with the optimism regularized objective,

$$\widetilde{f}_t \in \arg\min_{f \in \mathcal{F}} \frac{1}{|\mathcal{D}_t|} \sum_{(a,y) \in \mathcal{D}_t} (f(a) - y)^2 - \underbrace{\lambda_{\text{reg}} \mathbb{A}(f, \mathcal{U}_t)}_{\text{Optimism Regularizer}} \quad . \tag{3}$$

And define $B_t$ following Equation 2. Problem 3 is compatible with DNN function approximation. In our experiments we set the acquisition function to $\mathbb{A}_{\text{avg}}$, $\mathbb{A}_{\text{hinge}}$ with $p = 4$ and $\mathbb{A}_{\text{softmax}}$. The resulting methods are $\text{MeanOpt}$, $\text{HingePNorm}$ and $\text{MaxOpt}$. Throughout this work $\text{RandomBatch}$ corresponds to uniform arm selection and $\text{Greedy}$ to setting the optimism regularizer to $0$.

### 4.1.2 ENSEMBLE METHODS

We consider two distinct methods ($\text{Ensemble}$ and $\text{EnsembleNoiseY}$) to produce uncertainty estimations based on ensemble predictions. In both we fit $M$ models $\{\widehat{f}_t^i\}_{i=1}^M$ to $\mathcal{D}_t$ and define

$$\widetilde{f}_t^o(a) = \frac{\max_{i=1,\cdots,M} \widehat{f}_t^i(a) + \min_{i=1,\cdots,M} \widehat{f}_t^i(a)}{2}, \qquad \tilde{u}_t(a) = \max_{i=1,\cdots,M} \widehat{f}_t^i(a) - f_t^o(a).$$

So that $\widetilde{f}_t = \widetilde{f}_t^o + \tilde{u}_t$. We explore two distinct methods to produce $M$ different models $\{\widehat{f}_t^i\}_{i=1}^M$ fit to $\mathcal{D}_t$ that differ in the origin of the model noise. $\text{Ensemble}$ produces $M$ models resulting of independent random initialization of their model parameters.

The $\text{EnsembleNoiseY}$ method injects 'label noise' into the dataset responses. For all $i \in \{1, \cdots, M\}$ we build a dataset $\mathcal{D}_t^i = \{(a, y + \xi) \text{ for } (a, y) \in \mathcal{D}_t, \xi \sim \mathcal{N}(0, 1)\}$ where $\xi$ is an i.i.d. zero mean Gaussian random sample. The functions $\{\widehat{f}_t^i\}_{i=1}^M$ are defined as,

$$\widehat{f}_t^i \in \arg\min_{f \in \mathcal{F}} \sum_{(a,y) \in \mathcal{D}_t^i} (f(a) - y)^2.$$

In this case the uncertainty of the ensemble predictions is the result of both the random parameter initialization of the $\{\widehat{f}_t^i\}_{i=1}^M$ and the 'label noise'. This noise injection procedure draws its inspiration from methods such as RLSVI and NARL Russo (2019) and Pacchiano et al. (2021a).

## 4.2 DIVERSITY SEEKING VERSIONS OF OAE

In the case $b > 1$, the explore / exploit trade-off is not the sole consideration in selecting the arms that make up $B_t$. In this case, we should also be concerned about selecting sufficiently diverse points within the batch $B_t$ to maximize the information gathering ability of the batch. In Section 4.2 we show how to extend Algorithm 1 (henceforth referred to as vanilla OAE) to effectively induce

query diversity. We introduce two versions $\mathrm{OAE} - \mathrm{DvD}$ and $\mathrm{OAE} - \mathrm{Seq}$ which we discuss in more detail below. A detailed description of tractable implementations of these algorithms can be found in Appendix C.

### 4.2.1 DIVERSITY VIA DETERMINANTS

Inspired by diversity-seeking methods in the Determinantal Point Processes (DPPs) literature Kulesza & Taskar (2012), we introduce the $\mathrm{OAE} - \mathrm{DvD}$ algorithm. Inspired by the DvD algorithm Parker-Holder et al. (2020) we propose to augment the vanilla OAE objective with a diversity regularizer.

$$B_t \in \underset{\mathcal{B} \subseteq \mathcal{U}_t, |\mathcal{B}| = b}{\arg \max} \; \mathbb{A}_{\mathrm{sum}}(\widetilde{f}_t, \mathcal{B}) + \mathrm{Div}(\widetilde{f}_t, \mathcal{B}). \tag{4}$$

$\mathrm{OAE} - \mathrm{DvD}$'s Div regularizer is inspired by the theory of Determinantal Point Processes and equals a weighted log-determinant objective. $\mathrm{OAE} - \mathrm{DvD}$ has access to a kernel function $\mathcal{K} : \mathbb{R}^d \times \mathbb{R}^d \to \mathbb{R}$ and at the start of every time step $t \in \mathbb{N}$ it builds a kernel matrix $\mathbf{K}_t \in \mathbb{R}^{|\mathcal{U}_t| \times |\mathcal{U}_t|}$,

$$\mathbf{K}_t[i, j] = \mathcal{K}(a_i, a_j), \quad \forall i, j \in \mathcal{U}_t.$$

For any subset $\mathcal{B} \subseteq \mathcal{U}_t$ we define the $\mathrm{OAE} - \mathrm{DvD}$ diversity-aware score as,

$$\mathrm{Div}(f, \mathcal{B}) = \lambda_{\mathrm{div}} \log\left(\mathrm{Det}(\mathbf{K}_t[\mathcal{B}, \mathcal{B}])\right) \tag{5}$$

Where $\mathbf{K}_t[\mathcal{B}, \mathcal{B}]$ corresponds to the submatrix of $\mathbf{K}_t$ with columns (and rows) indexed by $\mathcal{B}$ and $\lambda_{\mathrm{div}}$ is a diversity regularizer. Since the resulting optimization problem in Equation 6 may prove to be extremely hard to solve, we design a greedy maximization algorithm to produce a surrogate solution. The details can be found in Appendix B.1. $\mathrm{OAE} - \mathrm{DvD}$ induces diversity leveraging the geometry of the action space.

### 4.2.2 SEQUENTIAL BATCH SELECTION RULES

In this section we introduce $\mathrm{OAE} - \mathrm{Seq}$ a generalization of the OAE algorithm designed to produce in batch diversity. $\mathrm{OAE} - \mathrm{Seq}$ produces a query batch by solving a sequence of $b$ optimization problems. The first element $a_{t,1}$ of batch $B_t$ is chosen as the arm in $\mathcal{U}_t$ achieving the most optimistic prediction over plausible models (following objective 1 and any of the tractable implementations defined in Section C, either optimistic regularization or ensemble methods). To produce the second point $a_{t,2}$ in the batch (provided $b > 1$) we temporarily add the pair $(a_{t,1}, \tilde{y}_{t,1})$ to the data buffer $\mathcal{D}_t$, where $\tilde{y}_{t,1}$ is a virtual reward estimator for $a_{t,1}$. Using this 'fake labels' augmented data-set we select $a_{t,2}$ following the same optimistic selection method used for $a_{t,1}$. Although other choices are possible in practice we set $\tilde{y}_{t,1}$ as a mid-point between an optimistic and pessimistic prediction of the value of $y_*(a_{t,1})$. The name $\mathrm{OAE} - \mathrm{Seq}$ derives from the 'sequential' way in which the batch is produced. If $\mathrm{OAE} - \mathrm{Seq}$ selected arm $a$ to be in the batch, and this arm has a non-optimistic virtual reward $\tilde{y}(a)$ that is low relative to the optimistic values of other arms, then $\mathrm{OAE} - \mathrm{Seq}$ will not select too many arms close to $a$ in the same batch. $\mathrm{OAE} - \mathrm{Seq}$ induces diversity not through the geometry of the arm space but in a way that is intimately related to the plausible arm values in the function class. A similar technique of adding hallucinated values to induce diversity has been proposed before, for example in the $\mathrm{GP} - \mathrm{BUCB}$ algorithm of Desautels et al. (2014). Ours is the first time this general idea has been tested in conjunction with scalable neural network function approximation algorithms. A detailed discussion of this algorithm can be found in Section B.1.1.

### 4.3 THE STATISTICAL COMPLEXITY OF ZERO NOISE BATCH LEARNING

In this section we present our main theoretical results regarding OAE with function approximation. In our results we use the Eluder dimension Russo & Van Roy (2013) to characterize the complexity of the function class $\mathcal{F}$. This is appropriate because our algorithms make use of the optimism principle to produce their queries $B_t$. We show two novel results. First, we characterize the sample complexity of zero noise parallel optimistic learning with Eluder classes with dimension $d$. Perhaps surprisingly the regret of Vanilla OAE with batch size $b$ has the same regret profile the case $b = 1$ up to a constant burn in factor of order $bd$. Second, our results holds under model misspecification, that is when $y_\star \notin \mathcal{F}$ at the cost of a linear dependence in the misspecification error. Although our results are for the noiseless setting (the subject of this work), we have laid the most important part of the groundwork

to extend them to the case of noisy evaluation. We explain why in Appendix D.2. We relegate the formal definition of the Eluder to Appendix D.2.

In this section we measure the misspecification of $y_\star$ via the $\|\cdot\|_\infty$ norm. We assume $y_\star$ satisfies $\min_{f \in \mathcal{F}} \|f - y_\star\|_\infty \leq \omega$ where $\|f - y_\star\|_\infty = \max_{a \in \mathcal{A}} |f(a) - y_\star(a)|$. Let $\widetilde{y}_\star = \arg\min_{f \in \mathcal{F}} \|y_\star - f\|_\infty$ be the $\|\cdot\|_\infty$ projection of $y_\star$ onto $\mathcal{F}$. We analyze OAE where $\widetilde{f}_t$ is computed by solving 1 with $\gamma_t \geq (t-1)b\omega^2$ with acquisition objective $\mathbb{A}_{\max,b}$. We will measure the performance of OAE via its regret defined as,

$$\text{Reg}_{\mathcal{F},b}(T) = \sum_{\ell=1}^{T} \left( \sum_{a \in B_{\star,t}} y_\star(a) - \sum_{a \in B_t} y_\star(a) \right).$$

Where $B_{\star,t} = \max_{\mathcal{B} \subset \mathcal{U}_t, |\mathcal{B}|=b} \sum_{a \in \mathcal{B}} y_\star(a)$. The main result in this section is,

**Theorem 4.1.** *The regret of* OAE *with* $\mathbb{A}_{\max,b}$ *acquisition function satisfies,*

$$\text{Reg}_{\mathcal{F},b}(T) = \mathcal{O}\left( \dim_E(\mathcal{F}, \alpha_T)b + \omega\sqrt{\dim_E(\mathcal{F}, \alpha_T)Tb} \right).$$

*With* $\alpha_t = \max\left(\frac{1}{t^2}, \inf\{\|f_1 - f_2\|_\infty : f_1, f_2 \in \mathcal{F}, f_1 \neq f_2\}\right)$ *and* $C = \max_{f \in \mathcal{F}, a, a' \in \mathcal{A}} |f(a) - f(a')|$.

The proof of theorem 4.1 can be found in Appendix D.2. This result implies the regret is bounded by a quantity that grows linearly with $\omega$, the amount of misspecification but otherwise only with the scale of $\dim_E(\mathcal{F}, \alpha_T)b$. The misspecification part of the regret scales as the same rate as a sequential algorithm running $Tb$ batches of size 1. When $\omega = 0$, the regret is upper bounded by $\mathcal{O}\left(\dim_E(\mathcal{F}, \alpha_T)d\right)$. For example, in the case of linear models, the authors of Russo & Van Roy (2013) show $\dim_E(\mathcal{F}, \epsilon) = \mathcal{O}\left(d \log(1/\epsilon)\right)$. This shows that for example sequential OAE when $b = 1$ achieves the lower bound of Lemma D.1 up to logarithmic factors. In the setting of linear models, the $b$ dependence in the rate above is unimprovable by vanilla OAE without diversity aware sample selection. This is because an optimistic algorithm may choose to use all samples in each batch to explore a single unexplored one dimensional direction. Theoretical analysis for $\text{OAE} - \text{DvD}$ and $\text{OAE} - \text{Seq}$ is left for future work. In Appendix D.1 we also show lower bounds for the query complexity for linear and Lipshitz classes.

## 5 TRANSFER LEARNING ACROSS GENETIC PERTURBATION DATASETS

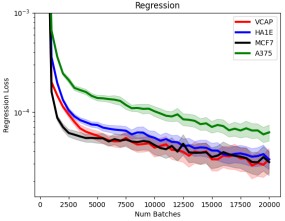

Figure 1: Regression mean squared error loss for models that predict cell-line specific phenotypic rewards from VCAP-derived perturbational features.

In order to show the effectiveness of OAE in the large batch - small number of iterations regime we consider genetic perturbations from the CMAP dataset Subramanian et al. (2017), which contains a 978-gene bulk expression readout from thousands of single-gene shRNA knockout perturbations[3] across a number of cell lines.

We consider the setting in which we have observed the effect of knockouts in one biological context (i.e., cell line) and would like to use it to plan a series of knockout experiments in another. Related applications may have different biological contexts, from different cell types or experimental conditions. We use the level 5 CMAP observations, each of which contains of 978-gene transcriptional readout from an shRNA knockout of a particular gene in a particular cell line. In our experiments, we choose to optimize a cellular proliferation phenotype, defined as a function on the 978-gene expression space. See Appendix F for details.

We use the 4 cells lines with the most number genetic perturbations in common: VCAP (prostate cancer, $n = 14602$ ), HA1E (kidney epithelium, $n = 10487$), MCF7 (breast cancer, $n = 6638$), and A375 (melanoma, $n = 10033$). We first learn a 100-dimensional action (perturbation) embedding $a_i$ for each perturbation in VCAP with an autoencoder. The autoencoder has a 100-dimension

---

[3]The shRNA perturbations are just a subset of the 1M+ total perturbations across different perturbation classes.

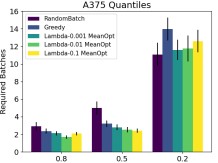 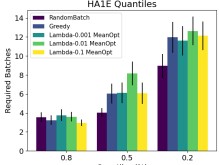 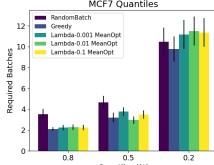 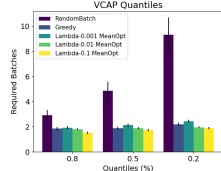

Figure 3: **Linear**. $\tau$-quantile batch convergence of MeanOpt on genetic perturbation datasets.

bottleneck layer and two intermediate layers of 1500 and 300 ReLU units with dropout and batch normalization and is trained using the Adam optimizer on mean squared reconstruction loss. We use these 100-dimensional perturbations embeddings as the features to train the $\widetilde{f}_t$ functions for each of the other cell types. According to our OAE algorithm, we train a fresh feed-forward neural network with two intermediate layers (of 100 and 10 units) for after observing the phenotypic rewards for each batch of 50 gene (knockout) perturbations. Figure 6 summarizes this approach.

Figure 1 shows the mean squared error loss of models trained to predict the cell-line specific phenotypic reward from the 100-dimensional VCAP-derived perturbational features. These models are trained on successive batches of perturbations sampled via RandomBatch and using the same **NN 1500-300** hidden layer architecture of the decoder. Not surprisingly, the loss for the VCAP reward is one of the lowest, but that of two other cell lines (HA1E and MCF7) are also quite similar, showing the **NN 1500-300** neural net function class is flexible to learn the reward function in one context from the perturbational embedding in another.

In all of our experiments we consider either a linear or a neural network function class with ReLU activations. In all of them we consider a batch size $b = 50$ and a number of batches $N = 20$. Figure 2 shows the convergence and reward results for the 4 cell lines when the neural network architecture equals **NN 100-10**. Since the perturbation action features were learned on the VCAP dataset (though agnostic to any phenotypic reward), the optimal VCAP perturbations are found quite quickly by all versions of MeanOpt including Greedy. Interestingly, MeanOpt still outperforms RandomBatch in the HA1E and MCF7 cell lines but

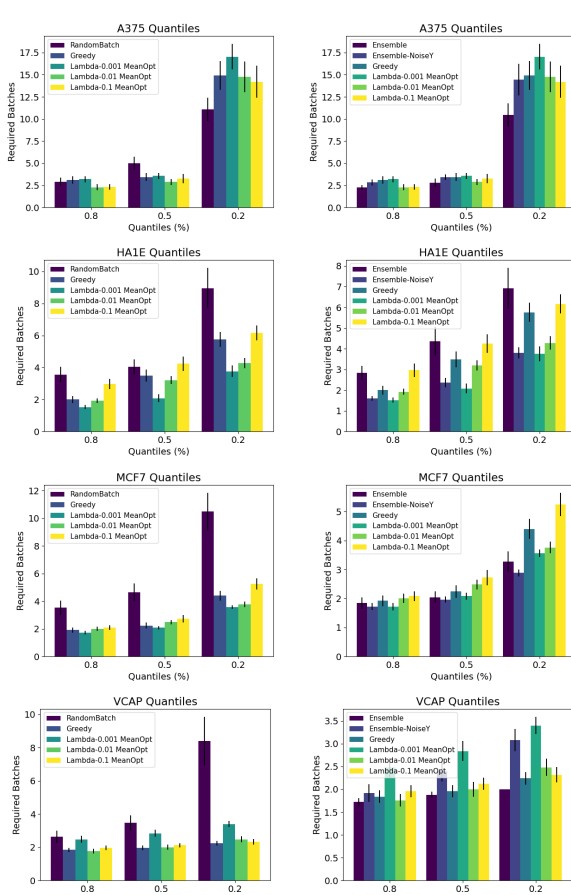

Figure 2: **NN 100-10** $\tau$-quantile batch convergence on genetic perturbation datasets of MeanOpt (**left**) and Ensemble, EnsembleNoiseY (**right**).

not on A375. When the neural network architecture equals **NN 10-5**, MeanOpt is only competitive with RandomBatch on the VCAP and MCF7 datasets (see figure 37 in Appendix G.4). Moreover, when $\mathcal{F}$ is a class of linear functions, MeanOpt can beat RandomBatch only in VCAP. This can be explained by looking at the regression fit plots of figure 4. The baseline loss value is the highest for A375 even for **NN 100-10**, thus indicating this function class is too far from the true responses values for A375. The loss curves for **NN 10-5** lie well above those for **NN 100-10** for all datasets thus explaining the degradation in performance when switching from a **NN 100-10** to a smaller capacity

of **NN 10-5**. Finally, the linear fit achieves a very small loss for VCAP explaining why MeanOpt still outperforms RandomBatch in VCAP with linear models. In all other datasets the linear fit is subpar to the **NN 100-10**, explaining why MeanOpt in **NN 100-10** works better than in linear ones. We note that EnsembleOptimism is competitive with RandomBatch in both **NN 10-5** and **NN 100-10** architectures in 7 our of the 8 experiments we conducted. In Appendix G.4 the reader can find results for MaxOpt and HingeOpt. Both methods underperform in comparison with MeanOpt. In Appendix E.2.1 and E.2.2 the reader will find experiments using tractable versions of $OAE - DvD$ and $OAE - Seq$. In Appendix E and G, we present extensive additional experiments and discussion of our findings over different network architectures (including linear), and over a variety of synthetic and public datasets from the UCI database Dua & Graff (2017).

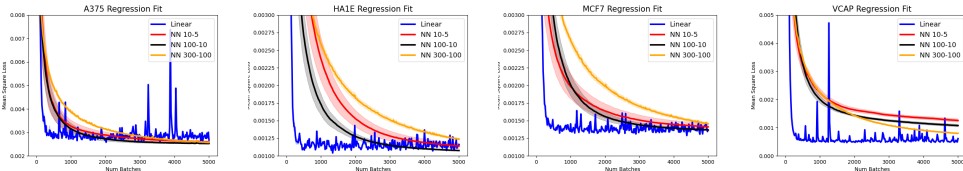

Figure 4: **Training set** regression fit curves evolution over training for the suite of gene perturbation datasets. Each training batch contains 10 datapoints.

### 5.0.1 GENEDISCO EXPERIMENTAL PLANNING BENCHMARK

We test a Bayesian OAE algorithm against the GeneDisco benchmark Mehrjou et al. (2021), which assess the "Hit Rate" of different experimental planning algorithms over a number of pooled CRISPR experiments. We assess our performance against the other acquisition functions provided in the public implementation[4] that select batches based solely on uncertainty considerations. We use the public implementation of GeneDisco and do not change the neural network architecture provided corresponding to a Bayesian Neural Network with a hidden layer of size 32. We use the 808 dimensional Achilles treatment descriptors and built $\widetilde{f_t}$ by adding the BNN's uncertainty to the model base predictions. We test our algorithm in the Schmidt et al. 2021 (IFNg), Schmidt et al. 2021 (IL2), Zhuang et al. 2019 (NK), and Zhu et al. 2021 (SarsCov2) datasets and tested for performance using the Hit Ratio metric after collecting 50 batches of size 16. This is defined as the ratio of arm pulls lying in the top .05 quantile of genes with the largest absolute value. Our results are in Table 5. OAE outperforms the other algorithms by a substantial margin in 3 out of the 4 datasets that we tested.

| Dataset | TopUncertain | SoftUncertain | OAE-Bayesian |
|---|---|---|---|
| Schmidt et al. 2021 (IFNg) | 0.057 | 0.046 | **0.062** |
| Schmidt et al. 2021 (IL2) | 0.083 | 0.081 | **0.107** |
| Zhuang et al. 2019 (NK) | 0.035 | 0.047 | **0.085** |
| Zhu et al. 2021 (SarsCov2) | 0.035 | **0.049** | 0.0411 |

Figure 5: Hit Ratio Results after 50 batches of size 16. BNN model trained with Achilles treatment descriptors. Final Hit Ratio average of 5 independent runs. TopUncertain selects the 16 points with the largest uncertainty values and SoftUncertain samples them using a softmax distribution.

## 6 CONCLUSION

In this work we introduce a variety of algorithms inspired in the OAE principle for noiseless batch bandit optimization. We also show lower bounds for the query complexity for linear and Lipshitz classes as well as a novel regret upper bound in terms of the Eluder dimension of the query class. Our theoretical results hold under misspecification. Through a variety of experiments in synthetic, public and biological data we show that the different incarnations of OAE we propose in this work can quickly search through a space of actions for the almost optimal ones. This work focused in the case where the responses are noiseless. Extending our methods and experimental evaluation to the case where the responses are noisy is an exciting avenue for future work.

---

[4] https://github.com/genedisco/genedisco-starter

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

CONTENTS

## A  TRANSFER LEARNING ACROSS GENETIC PERTURBATION DATASETS

In this section we have placed a diagrammatic version of the data pipeline described in section 5.

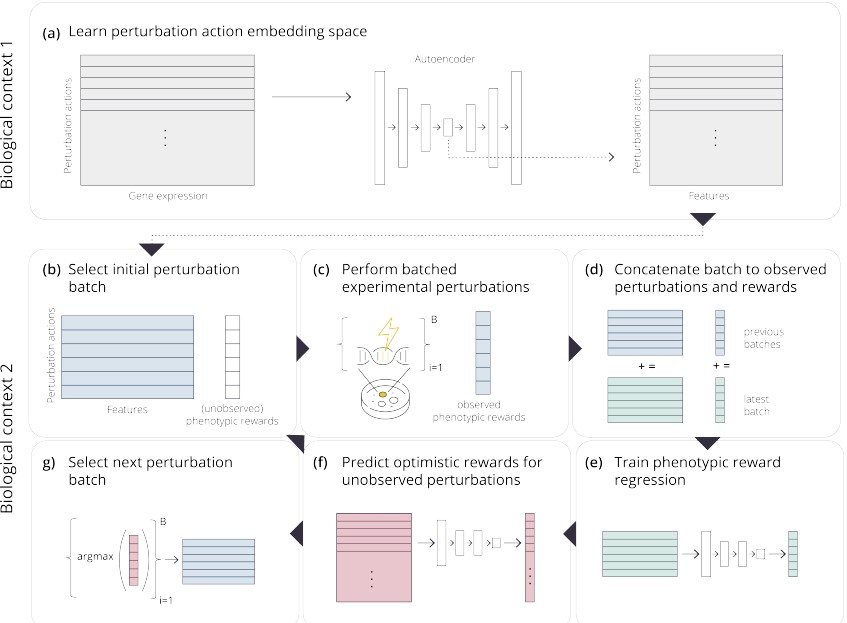

Figure 6:  Neural design for genetic perturbation experiments. (a) Learn a perturbation action embedding space by training an autoencoder on the gene expression resulting from a large set of observed genetic perturbations in a particular biological context (e.g., shRNA gene knockouts for a particular cell line in CMAP). (b) Select an initial batch of $B$ perturbation actions to perform in parallel within a related (but different) biological context. Selection can be random (uniform) or influenced by prior information about the relationship between genes and the phenotype to be optimized. (c) Perform the current batch of experimental perturbations *in vitro* and observed their corresponding phenotypic rewards. (d) Concatenate the latest batch's features and observed rewards to those of previous batches to update the perturbation reward training set. (e) Train a new perturbation reward regression (with some degree of pre-defined optimism) network on the observed perturbation rewards. (f) Use this regressor to predict the optimistic rewards for the currently unobserved perturbations. (g) Select the next batch from these unobserved perturbations with the highest optimistic reward.

# B COMPLEMENTARY DESCRIPTION OF THE METHODS

## B.1 APPENDIX DIVERSITY VIA DETERMINANTS

Inspired by diversity-seeking methods in the Determinantal Point Processes (DPPs) literature Kulesza & Taskar (2012), we introduce the $\mathrm{OAE} - \mathrm{DvD}$ algorithm. Inspired by the DvD algorithm **??** we propose the use

DPPs can be used to produces diverse subsets by sampling proportionally to the determinant of the kernel matrix of points within the subset. From a geometric perspective, the determinant of the kernel matrix represents the volume of a parallelepiped spanned by feature maps corresponding to the kernel choice. We seek to maximize this volume, effectively "filling" the feature space. Using a determinant score to induce diversity has been proposed as a strategy in other domains, most notably in the form of the Diversity via Determinants (DvD) algorithm from Parker-Holder et al. (2020) for Population Based Reinforcement Learning. It is from this work that we take inspiration to name our algorithm $\mathrm{OAE} - \mathrm{DvD}$. The idea of using DPPs for diversity guided active learning has been explored by Bıyık et al. (2019). In the active learning setting the objective function used to build the batch is purely driven by the diversity objective. The method works as follows. At time $t$, $\mathrm{OAE} - \mathrm{DvD}$ constructs a regression estimator $\widetilde{f}_t$ using the arms and responses in $\mathcal{D}_t$ (for example by solving problem 1). Instead of using Equation 2, our algorithm selects batch $\mathcal{B}_t$ by optimizing a diversity aware objective of the form,

$$B_t \in \underset{\mathcal{B} \subseteq \mathcal{U}_t, |\mathcal{B}|=b}{\arg\max} \; \mathbb{A}_{\mathrm{sum}}(\widetilde{f}_t, \mathcal{B}) + \mathrm{Div}(\widetilde{f}_t, \mathcal{B}). \tag{6}$$

$\mathrm{OAE} - \mathrm{DvD}$'s Div regularizer is inspired by the theory of Determinantal Point Processes and equals a weighted log-determinant objective. $\mathrm{OAE} - \mathrm{DvD}$ has access to a kernel function $\mathcal{K} : \mathbb{R}^d \times \mathbb{R}^d \to \mathbb{R}$ and at the start of every time step $t \in \mathbb{N}$ it builds a kernel matrix $\mathbf{K}_t \in \mathbb{R}^{|\mathcal{U}_t| \times |\mathcal{U}_t|}$,

$$\mathbf{K}_t[i, j] = \mathcal{K}(a_i, a_j), \quad \forall i, j \in \mathcal{U}_t.$$

For any subset $\mathcal{B} \subseteq \mathcal{U}_t$ we define the $\mathrm{OAE} - \mathrm{DvD}$ diversity-aware score as,

$$\mathrm{Div}(f, \mathcal{B}) = \lambda_{\mathrm{div}} \log\left(\mathrm{Det}(\mathbf{K}_t[\mathcal{B}, \mathcal{B}])\right) \tag{7}$$

Where $\mathbf{K}_t[\mathcal{B}, \mathcal{B}]$ corresponds to the submatrix of $\mathbf{K}_t$ with columns (and rows) indexed by $\mathcal{B}$ and $\lambda_{\mathrm{div}}$ is a diversity regularizer. Since the resulting optimization problem in Equation 6 may prove to be extremely hard to solve, we design a greedy maximization algorithm to produce a surrogate solution. We build the batch $B_t$ greedily. The first point $a_{t,1}$ in the batch is selected to be the point in $\mathcal{U}_t$ that maximizes the response $\widetilde{f}_t$. For all $i \geq 2$ the point $a_{t,i}$ in $\mathcal{U}_t$ is selected from $\mathcal{U}_t \backslash \{a_{t,j}\}_{j=1}^{i-1}$ such that,

$$a_{t,i} = \max_{a \in \mathcal{U}_t \backslash \{a_{t,j}\}_{j=1}^{i-1}} \mathbb{A}_{\mathrm{sum}}(\widetilde{f}_t, \{a\} \cup \{a_{t,j}\}_{j=1}^{i-1}) + \mathrm{Div}(\widetilde{f}_t, \{a\} \cup \{a_{t,j}\}_{j=1}^{i-1})$$

---

**Algorithm 2** Optimistic Arm Elimination - DvD ($\mathrm{OAE} - \mathrm{DvD}$)

---

**Input** Action set $\mathcal{A} \subset \mathbb{R}^d$, num batches $N$, batch size $b$, $\lambda_{\mathrm{div}}$
**Initialize** Unpulled arms $\mathcal{U}_1 = \mathcal{A}$. Observed points and labels dataset $\mathcal{D}_1 = \emptyset$
**for** $t = 1, \cdots, N$ **do**
    **if** $t = 1$ **then**
        Sample uniformly a size $b$ batch $B_t \sim \mathcal{U}_1$.
    **else**
        Compute $B_t$ using the greedy procedure described above.

    Observe batch rewards $Y_t = \{y_*(a) \text{ for } a \in B_t\}$
    Update $\mathcal{D}_{t+1} = \mathcal{D}_t \cup \{(B_t, Y_t)\}$.
    Update $\mathcal{U}_{t+1} = \mathcal{U}_t \backslash B_t$.

---

We define a reward augmented kernel matrix $\mathbf{K}_t^{\mathrm{aug}}(i, j) = \mathbf{K}_t(i, j)\sqrt{\exp\left(\frac{\widetilde{f}_t(i) + \widetilde{f}_t(j)}{\lambda_{\mathrm{det}}}\right)}$. This matrix

satisfies $\mathbf{K}_t^{\mathrm{aug}}[\mathcal{B}, \mathcal{B}] = \mathrm{diag}\left(\sqrt{\exp(\widetilde{f}_t/\lambda_{\mathrm{det}})}\right)[\mathcal{B}, \mathcal{B}] \cdot \mathbf{K}_t[\mathcal{B}, \mathcal{B}] \cdot \mathrm{diag}\left(\sqrt{\exp(\widetilde{f}_t/\lambda_{\mathrm{det}})}\right)[\mathcal{B}, \mathcal{B}]$

for all $\mathcal{B} \subseteq \mathcal{U}_t$. Since the determinant of a product of matrices is the product of their determinants it follows that $\mathrm{Det}(\mathbf{K}_t^{\mathrm{aug}}[\mathcal{B}, \mathcal{B}]) = \exp\left(\sum_{a \in \mathcal{B}} \widetilde{f}_t(a)/\lambda_{\mathrm{det}}\right) \cdot \mathrm{Det}\left(\mathbf{K}_t[\mathcal{B}, \mathcal{B}]\right)$. Thus for all $\mathcal{B} \subseteq \mathcal{U}_t$, equation 6 with diversity score 7 can be rewritten as

$$B_t \in \underset{\mathcal{B} \subseteq \mathcal{U}_t, |\mathcal{B}|=b}{\arg\max} \; \log\left(\mathrm{Det}(\mathbf{K}_t^{\mathrm{aug}}[\mathcal{B}, \mathcal{B}])\right)$$

This is because $\log\left(\mathrm{Det}(\mathbf{K}_t^{\mathrm{aug}}[\mathcal{B}, \mathcal{B}])\right) = \frac{\sum_{a \in \mathcal{B}} \widetilde{f}_t(a)}{\lambda_{\mathrm{div}}} + \log\left(\mathrm{Det}(\mathbf{K}_t[\mathcal{B}, \mathcal{B}])\right)$ for all $\mathcal{B} \subset \mathcal{U}_t$. It is well known the log-determinant set function for a positive semidefinite matrix is submodular (see for example section 2.2 of Han et al. (2017)). It has long been established that the greedy algorithm achieves an approximation ratio of $(1 - \frac{1}{e})$ for the constrained submodular optimization problem (see Nemhauser et al. (1978)). This justifies the choices we have made behind the greedy algorithm we use to select $B_t$.

### B.1.1 SEQUENTIAL BATCH SELECTION RULES

In this section we introduce $\mathrm{OAE} - \mathrm{Seq}$ a generalization of the OAE algorithm designed to produce in batch diversity. $\mathrm{OAE} - \mathrm{Seq}$ produces a query batch by solving a sequence of $b$ optimization problems. $\mathrm{OAE} - \mathrm{Seq}$ uses first $\widetilde{\mathcal{D}}_{t,0} = \mathcal{D}_t$ the set of arms pulled so far as well as $\widetilde{\mathcal{U}}_{t,0} = \mathcal{U}_t$ (the set of arms yet to be pulled) to produce a function $\widetilde{f}_{t,1}$ that determines the initial arm in the batch via the greedy choice $a_{t,1} = \arg\max_{a \in \mathcal{U}_t} \widetilde{f}_{t,1}(a)$. The function $\widetilde{f}_{t,1}$ is computed using the same method as any vanilla OAE procedure. Having chosen this arm, in the case when $b > 1$, a virtual reward $\widetilde{y}_{t,1}$ (possibly different from $\widetilde{f}_t(a_{t,1})$) is assigned to the query arm $a_{t,1}$, and datasets $\widetilde{\mathcal{D}}_{t,1} = \mathcal{D}_t \cup \{(a_{t,1}, \widetilde{y}_{t,1})\}$ and $\widetilde{\mathcal{U}}_{t,1} = \mathcal{U}_t \backslash \{a_{t,1}\}$ are defined. The same optimization procedure that produced $\widetilde{f}_{t,1}$ is used to output $\widetilde{f}_{t,2}$ now with $\widetilde{\mathcal{D}}_{t,1}$ and $\widetilde{\mathcal{U}}_{t,1}$ as inputs. Arm $a_{t,2}$ is defined as the greedy choice $a_{t,2} = \arg\max_{a \in \widetilde{\mathcal{U}}_{t,1}} \widetilde{f}_{t,2}(a)$. The remaining batch elements (if any) are determined by successive repetition of this process so that $\widetilde{D}_{t,i} = \mathcal{D}_t \cup \{(a_{t,j}, \widetilde{y}_{t,j})\}_{j=1}^i$ and $\widetilde{\mathcal{U}}_{t,i} = \mathcal{U}_t \backslash \{a_{t,j}\}_{j=1}^i$. The trace of this procedure leaves behind a sequence of functions and datasets $\{(\widetilde{f}_{t,i}, \widetilde{\mathcal{U}}_{t,i})\}_{i=1}^b$ such that $a_{t,i} \in \arg\max_{\widetilde{\mathcal{U}}_{t,i-1}} \widetilde{f}_{t,i}(a)$.

---

**Algorithm 3** Optimistic Arm Elimination - Batch Sequential ($\mathrm{OAE} - \mathrm{Seq}$)

---

**Input** Action set $\mathcal{A} \subset \mathbb{R}^d$, number of batches $N$, batch size $b$, pessimism-optimism balancing parameter $\alpha$.
**Initialize** Unpulled arms $\mathcal{U}_1 = \mathcal{A}$. Observed points and labels dataset $\mathcal{D}_1 = \emptyset$
**for** $t = 1, \cdots, N$ **do**

    **if** $t = 1$ **then**
        Sample uniformly a size $b$ batch $B_t \sim \mathcal{U}_1$.

    **else**
        Solve for $\{(\widetilde{f}_{t,i}, \widetilde{\mathcal{U}}_{t,i})\}_{i=1}^b$ via the procedure described in the text.
        Compute

$$B_t = \{a_{t,i} \in \underset{a \in \widetilde{U}_{t,i-1}}{\arg\max} \; \widetilde{f}_{t,i}(a)\}_{i=1}^b. \tag{8}$$

    Observe batch rewards $Y_t = \{f_*(a) \text{ for } a \in B_t\}$
    Update $\mathcal{D}_{t+1} = \mathcal{D}_t \cup \{(B_t, Y_t)\} \in \mathcal{D}$.
    Update $\mathcal{U}_{t+1} = \mathcal{U}_t \backslash B_t$.

---

To determine the value of the virtual rewards $\widetilde{y}_{t,i}$, we consider a variety of options. We start by discussing the case when the fake reward $\widetilde{y}_{t,i} = \widetilde{f}_t(a_{t,i})$ and the acquisition function equals $\mathbb{A}_{\mathrm{sum}}(f, \mathcal{U})$.

When $\gamma_t = 0$, the fake reward satisfies $\widetilde{y}_{t,i} = \widetilde{f}_t(a_{t,i})$ and $y_\star \in \mathcal{F}$ it follows that $\widetilde{f}_{t,i} = \widetilde{f}_t$ independent of $i \in [B]$ is a valid choice for the function ensemble $\{\widetilde{f}_{t,i}\}_{i=1}^b$. In this case the query batch $B_t$ can be computed by solving for $\widetilde{f}_t$,

$$\widetilde{f}_t \in \arg\max_{f \in \mathcal{F}} \sum_{a \in \mathcal{U}_t} f(a) \qquad \text{s.t.} \sum_{(a,y) \in \widetilde{\mathcal{D}}_{t,i}} (f(a) - y)^2 = 0. \tag{9}$$

And defining the batch $B_t$ as

$$B_t = \arg\max_{\mathcal{B} \subseteq \mathcal{U}_t \text{ s.t. } |\mathcal{B}|=b} \sum_{a \in \mathcal{B}} \widetilde{f}_t(a).$$

The equivalence between this definition of $B_t$ and the sequential batch selection rule follows by noting the equality constraint from 9 ensures that $\widetilde{f}_{t,i} = \widetilde{f}_t$ is a valid solution for each of the intermediate problems defining the sequence $\{(\widetilde{f}_{t,i}, \widetilde{U}_{t,i})\}_{i=1}^B$.

$\mathrm{OAE} - \mathrm{Seq}$ is designed with a more general batch selection rule that may yield distinct intermediate arm selection functions $\{\widetilde{f}_{t,i}\}$. In our experiments we compute the virtual reward $\widetilde{y}_{t,i}$ as an average of $\widetilde{f}_{t,i}^{\mathrm{optimistic}}$ and $\widetilde{f}_{t,i}^{\mathrm{pessimistic}}$, optimistic and a pessimistic estimators of the responses in $\mathcal{U}_{t,i-1}$.

We consider two mechanisms for computing $\widetilde{f}_{t,i}^{\mathrm{optimistic}}$ and $\widetilde{f}_{t,i}^{\mathrm{pessimistic}}$. First when the computation of $\widetilde{f}_{t,i}$ is based on producing an uncertainty function $\tilde{u}_{t,i}$ and a base model $\widetilde{f}_{t,i}^o$, we define $\widetilde{f}_{t,i}^{\mathrm{optimistic}} = \widetilde{f}_{t,i}^o + u_{t,i}$ and $\widetilde{f}_{t,i}^{\mathrm{pessimistic}} = \widetilde{f}_{t,i}^o - \tilde{u}_{t,i}$. Second, we define $\widetilde{f}_{t,i}^{\mathrm{optimistic}}$ and $\widetilde{f}_{t,i}^{\mathrm{pessimistic}}$ as the solutions of the constrained objectives

$$\widetilde{f}_{t,i}^{\mathrm{optimistic}} = \arg\max_{f \in \mathcal{F}_{t,i}} \mathbb{A}(f, \mathcal{U}_t) \qquad \text{and} \qquad \widetilde{f}_{t,i}^{\mathrm{pessimistic}} = \arg\min_{f \in \mathcal{F}_{t,i}} \mathbb{A}(f, \mathcal{U}_t)$$

Where $\mathcal{F}_{t,i} = \{f \in \mathcal{F} \text{ s.t. } \sum_{(a,y) \in \mathcal{D}_{t,i-1}} (f(a) - y)^2 \le \gamma_t\}$. In both cases we define the fictitious rewards as a weighted average of the pessimistic and optimistic predictors $\widetilde{y}_{t,i} = \alpha \widetilde{f}_{t,i}^{\mathrm{optimistic}} + (1 - \alpha)\widetilde{f}_{t,i}^{\mathrm{pessimistic}}$ where $\alpha \in [0, 1]$ is an optimism weighting parameter while we keep the functions used to define what points are part of the batch as $\widetilde{f}_{t,i} = \widetilde{f}_{t,i}^{\mathrm{optimistic}}$. The principle of adding hallucinated values to induce diversity has been proposed before for example in the $\mathrm{GP} - \mathrm{BUCB}$ algorithm of Desautels et al. (2014).

## C  TRACTABLE IMPLEMENTATIONS OF $\mathrm{OAE} - \mathrm{DvD}$ AND $\mathrm{OAE} - \mathrm{Seq}$

In this section we go over the algorithmic details behind the approximations that we have used when implementing the different OAE methods we have introduced in Section 4.

### C.0.1  DIVERSITY VIA DETERMINANTS

The $\mathrm{OAE} - \mathrm{DvD}$ algorithm differs from OAE only in the way in which the query batch $B_t$ is computed. $\mathrm{OAE} - \mathrm{DvD}$ uses equation 6 instead of equation 2. Solving for $B_t$ is done via the greedy algorithm described in Section 4.2.1. The function $\widetilde{f}_t$ can be computed via a regularized optimization objective or using an ensemble. In our experimental evaluation we opt for defining $\widetilde{f}_t$ via the regularization route as the result of solving problem 3. More experimental details including the type of kernel used are explained in Section E.2. In our experimental evaluation we use the $\mathrm{MeanOpt}$ objective to produce $\widetilde{f}_t$ and we refer to the resulting method as $\mathrm{DetD}$.

### C.0.2  SEQUENTIAL BATCH SELECTION RULES

We explore different ways of defining the functions $\{\widetilde{f}_{t,i}\}_{i=1}^b$. Depending on what procedure we use to optimize and produce these functions we will obtain different versions of $\mathrm{OAE} - \mathrm{Seq}$. We use the name SeqB to denote the $\mathrm{OAE} - \mathrm{Seq}$ method that fits the functions $\{\widetilde{f}_{t,i}^{\mathrm{optimistic}}\}_{i=1}^b$ and

$\{\widetilde{f}_{t,i}^{\text{pessimistic}}\}_{i=1}^{b}$ by solving the regularized objectives,

$$\widetilde{f}_{t,i}^{\text{optimistic}} \in \arg\min_{f\in\mathcal{F}} \frac{1}{|\mathcal{D}_{t,i-1}|} \sum_{(a,y)\in\mathcal{D}_{t,i-1}} (f(a)-y)^2 - \lambda_{\text{reg}}\mathbb{A}(f,\mathcal{U}_{t,i-1}),$$

$$\widetilde{f}_{t,i}^{\text{pessimistic}} \in \arg\min_{f\in\mathcal{F}} \frac{1}{|\mathcal{D}_{t,i-1}|} \sum_{(a,y)\in\mathcal{D}_{t,i-1}} (f(a)-y)^2 + \lambda_{\text{reg}}\mathbb{A}(f,\mathcal{U}_{t,i-1}).$$

In our experiments we set the acquisition function to $\mathbb{A}_{\text{avg}}$. For some value[5] of $\lambda_{\text{reg}} > 0$. The functions $\{\widetilde{f}_{t,i}\}_{i=1}^{b}$ are defined as $\widetilde{f}_{t,i} = \widetilde{f}_{t,i}^{\text{optimistic}}$ and the virtual rewards as $\widetilde{y}_{t,i} = \alpha\widetilde{f}_{t,i}^{\text{optimistic}} + (1-\alpha)\widetilde{f}_{t,i}^{\text{pessimistic}}$ where $\alpha \in [0,1]$ is an optimism-pessimism weighting parameter. In our experimental evaluation we use $\alpha = \frac{1}{2}$. More experimental details are presented in Section E.2.

The functions $\{\widetilde{f}_{t,i}^{\text{optimistic}}\}_{i=1}^{b}$ and $\{\widetilde{f}_{t,i}^{\text{pessimistic}}\}_{i=1}^{b}$ can be defined with the use of an ensemble. Borrowing the definitions of Section 4.1.2

$$\widetilde{f}_{t,i}^{\text{optimistic}} = \widetilde{f}_{t,i}^{o} + \tilde{u}_{t,i}, \qquad \widetilde{f}_{t,i}^{\text{pessimistic}} = \widetilde{f}_{t,i}^{o} - \tilde{u}_{t,i}.$$

Where $\widetilde{f}_{t,i}^{o}$ and $\tilde{u}_{t,i}$ are computed by first fitting an ensemble of models $\{\widehat{f}_{t,i}^{j}\}_{j=1}^{M}$ using dataset $\mathcal{D}_{t,i-1}$. In our experimental evaluation we explore the use of Ensemble and EnsembleNoiseY optimization styles to fit the models $\{\widehat{f}_{t,i}^{j}\}_{j=1}^{M}$. In our experiments we use the names $\text{Ensemble} - \text{SeqB}$ and $\text{Ensemble} - \text{SeqB} - \text{NoiseY}$ to denote the resulting sequential batch selection methods. More details of our implementation can be found in Section E.2.

## D  THEORETICAL RESULTS

### D.1  QUANTIFYING THE QUERY COMPLEXITY OF $\mathcal{F}$

Let $\epsilon \geq 0$ and define $t_{\text{opt}}^{\epsilon}(\mathcal{A}, f)$ to be the first time-step when an $\epsilon$−optimal point $\hat{a}_t$ is proposed by a learner (possibly randomized) when interacting with arm set $\mathcal{A} \subset \mathbb{R}^d$ and the pseudo rewards are noiseless evaluations $\{f(a)\}_{a\in\mathcal{A}}$ with $f \in \mathcal{F}$. We define the query complexity of the $\mathcal{A}, \mathcal{F}$ pair as,

$$\mathbb{T}^{\epsilon}(\mathcal{A}, \mathcal{F}) = \min_{\text{Alg}} \max_{f\in\mathcal{F}} \mathbb{E}\left[t_{\text{opt}}^{\epsilon}(\mathcal{A}, f)\right]$$

Where the minimum iterates over all possible learning algorithms. We can lower bound of the problem complexity for several simple problem classes,

**Lemma D.1.** *When $\mathcal{A} = \{\|x\| \leq 1 \text{ for } x \in \mathbb{R}^d\}$ and*

1. *If $\mathcal{F}$ is the class of linear functions defined by vectors in the unit ball $\mathcal{F} = \{x \to \theta^\top x : \|\theta\| \leq 1 \text{ for } \theta \in \mathbb{R}^d\}$ then $\mathbb{T}^{\epsilon}(\mathcal{A}, \mathcal{F}) \geq d$ when $\epsilon < \frac{1}{d}$ and $\mathbb{T}^{\epsilon}(\mathcal{A}, \mathcal{F}) \geq \lceil d - \epsilon d \rceil$ otherwise.*

2. *If $\mathcal{F}$ is the class of 1-Lipschitz functions functions then $\mathbb{T}^{\epsilon}(\mathcal{A}, \mathcal{F}) \geq \left(\frac{1}{4\epsilon}\right)^d$.*

*Proof.* As a consequence of Yao's principle, we can restrict ourselves to deterministic algorithms. Indeed,

$$\min_{\text{Alg}} \max_{f\in\mathcal{F}} \mathbb{E}\left[t_{\text{opt}}^{\epsilon}(\mathcal{A}, f)\right] = \max_{\mathcal{D}_{\mathcal{F}}} \min_{\text{DetAlg}} \mathbb{E}_{f\sim\mathcal{D}_{\mathcal{F}}}\left[t_{\text{opt}}^{\epsilon}(\mathcal{A}, f)\right]$$

Thus, to prove the lower bound we are after it is enough to exhibit a distribution $\mathcal{D}_{\mathcal{F}}$ over instances $f \in \mathcal{F}$ and show a lower bound for the expected $t_{\text{opt}}^{\epsilon}(\mathcal{A}, f)$ where the expectation is taken using $\mathcal{D}_{\mathcal{F}}$.

With the objective of proving item 1 let $\mathcal{D}_{\mathcal{F}}$ be the uniform distribution over the sphere $\mathbf{Unif}_d(1)$. By symmetry it is easy to see that

$$\mathbb{E}_{\theta\sim\mathbf{Unif}_d(r)}[\theta_i] = 0, \quad \forall i \in d \text{ and } \forall r \geq 0.$$

---

[5]As we have pointed out in Section 4.2.2 setting $\lambda_{\text{reg}} = 0$ reduces $\text{OAE} - \text{Seq}$ to vanilla OAE.

Thus,

$$\text{Var}_{\theta\sim\mathbf{Unif}_d(r)}(\theta_i) = \mathbb{E}_{\theta\sim\mathbf{Unif}_d(r)}[\theta_i^2] \stackrel{(i)}{=} \frac{r^2}{d}. \tag{10}$$

Equality $(i)$ follows because

$$\sum_{i=1}^{d}\mathbb{E}_{\theta\sim\mathbf{Unif}_d(r)}[\theta_i^2] = \mathbb{E}_{\theta\sim\mathbf{Unif}_d(r)}[\|\theta\|^2] = r^2$$

and because by symmetry for all $i, j \in [d]$ the second moments agree,

$$\mathbb{E}_{\theta\sim\mathbf{Unif}_d(r)}[\theta_i^2] = \mathbb{E}_{\theta\sim\mathbf{Unif}_d(r)}[\theta_j^2]$$

Finally,

$$\mathbb{E}_{\theta\sim\mathbf{Unif}_d(r)}[|\theta_i|] \leq \sqrt{\mathbb{E}_{\theta\sim\mathbf{Unif}_d(r)}[\theta_i^2]} = \frac{r}{\sqrt{d}}. \tag{11}$$

Let $\mathrm{DetAlg}$ be the optimal deterministic algorithm for $\mathcal{D}_{\mathcal{F}}$ and $a_1$ be its first action. Since $\mathcal{D}_{\mathcal{F}}$ is the unofrm distribution over the sphere, inequality 11 expected scale of the reward reward experienced is upper bounded by $\frac{1}{\sqrt{d}}$, and furthermore, since $\|a_1\| = 1$, the expected second moment of the reward experienced (where expectations are taken over $\mathcal{D}_{\mathcal{F}}$) equals $\frac{1}{d}$.

We now employ a conditional argument, if $\mathrm{DetAlg}$ has played $a_1$ and observed a reward $r_1$,

We assume that up to time $m$ algorithm $\mathrm{DetAlg}$ has played actions $a_1, \cdots, a_m$ and received rewards $r_1, \cdots, r_m$.

Given these outcomes, $\mathrm{DetAlg}$ can recover the component of $\theta$ lying in $\mathbf{span}(a_1, \cdots, a_m)$. Let $a_{m+1}$ be $\mathrm{DetAlg}$'s action at time $m+1$. By assumption this is a deterministic function of $a_1, \cdots, a_m$ and $r_1, \cdots, r_m$. Since $\theta$ is drawn from $\mathbf{Unif}_d(1)$, the expected squared dot product between the component of $a_{m+1}^{\perp} = \mathbf{Proj}(a_{m+1}, \mathbf{span}(a_1, \cdots, a_m)^{\perp})$ satisfies,

$$\mathbb{E}_{\theta\sim\mathcal{D}_{\mathcal{F}}|\{a_1^{\top}\theta=r_i\}_{i=1}^{m}}\left[\left(\theta^{\top}a_{m+1}^{\perp}\right)^2\right] = \frac{1-\|\theta_m^0\|^2}{d-m}\left(1-\|a_{m+1}^0\|^2\right)$$
$$= \frac{1-\|\theta_m^0\|^2}{d-m}. \tag{12}$$

where $\theta_m^0 = \mathbf{Proj}(\theta, \mathbf{span}(a_1, \cdots, a_m))$. The last inequality follows because the conditional distribution of $\mathbf{Proj}(\theta, \mathbf{span}(a_1, \cdots, a_m)^{\perp})$ given $a_1, \cdots, a_m$ and $r_1, \cdots, r_m$ is a uniform distribution over the $d - m$ dimensional sphere of radius $\sqrt{1 - \|\theta_m^0\|^2}$, the scale of $a_{m+1}^{\perp}$ is $\sqrt{1 - \|a_{m+1}^0\|^2}$ and we have assumed the $\|a_{m+1}^0\| = 0$. Thus, the agreement of $a_{m+1}^{\perp}$ with $\mathbf{Proj}(\theta, \mathbf{span}(a_1, \cdots, a_m)^{\perp})$ satisfies Equation 10.

We consider the expected square norm of the recovered $\theta$ up to time $m$. This is the random variable $\|\theta_m^0\|^2 = \sum_{t=1}^{m}\left(\theta^{\top}a_t^{\perp}\right)^2$ where $a_t^{\perp} = \mathbf{Proj}(a_t, \mathbf{span}(a_1, \cdots, a_{t-1})^{\perp})$. Thus,

$$\mathbb{E}_{\theta\sim\mathcal{D}_{\mathcal{F}}}\left[\|\theta_m^0\|^2\right] = \mathbb{E}_{\theta\sim\mathcal{D}_{\mathcal{F}}}\left[\sum_{t=1}^{m}\left(\theta^{\top}a_t^{\perp}\right)^2\right]$$
$$= \mathbb{E}_{\theta\sim\mathcal{D}_{\mathcal{F}}}\left[\sum_{t=1}^{m}\mathbb{E}_{\theta\sim\mathcal{D}_{\mathcal{F}}|\{a_1^{\top}\theta=r_i\}_{i=1}^{t-1}}\left[\left(\theta^{\top}a_t^{\perp}\right)^2\right]\right]$$
$$\stackrel{(i)}{=} \mathbb{E}_{\theta\sim\mathcal{D}_{\mathcal{F}}}\left[\sum_{t=1}^{m}\frac{1-\|\theta_{t-1}^0\|^2}{d-m}\right]$$

Equality $(i)$ holds because of 12. Recall that by Equation 10,

$$\mathbb{E}_{\theta\sim\mathcal{D}_{\mathcal{F}}}\left[\|\theta_1^0\|^2\right] = \frac{1}{d}.$$

Thus by the above equalities,

$$\mathbb{E}_{\theta \sim \mathcal{D}_{\mathcal{F}}} \left[ \|\theta_2^0\|^2 \right] = \frac{1}{d} + \frac{1 - \frac{1}{d}}{d-1} = \frac{2}{d}.$$

Unrolling these equalities further we conclude that

$$\mathbb{E}_{\theta \sim \mathcal{D}_{\mathcal{F}}} \left[ \|\theta_m^0\|^2 \right] = \frac{m}{d}.$$

This implies the expected square agreement between the learner's virtual guess $\hat{a}_t$ is upper bounded by $\frac{m}{d}$. Thus, when $\epsilon < \frac{1}{d}$, the expected number of queries required is at least $d$. When $\epsilon > d$, the expected number of queries instead satisfies a lower bound of $\lceil d - \epsilon d \rceil$.

We now show shift our attention to Lipschitz functions. First we introduce the following simple construction of a $1-$Lipschitz function over a small ball of radius $\epsilon$. We use this construction throughout our proof. Let $\mathbf{x} \in \mathbb{R}^d$ be an arbitrary vector, define $B(\mathbf{x}, \epsilon)$ as the ball centered around $\mathbf{x}$ of radius $2\epsilon$ under the $\| \cdot \|_2$ norm and $S(\mathbf{x}, 2\epsilon)$ as the sphere (the surface of $B(\mathbf{x}, 2\epsilon)$) centered around $\mathbf{x}$

Define the function $f_{\mathbf{x}}^\epsilon : \mathbb{R}^d \to \mathbb{R}$ as,

$$f_{\mathbf{x}}^\epsilon(\mathbf{z}) = \begin{cases} \min_{\mathbf{z}' \in S(\mathbf{x}, 2\epsilon)} \|\mathbf{z} - \mathbf{z}'\|_2 & \text{if } \mathbf{z} \in B(\mathbf{x}, 2\epsilon) \\ 0 & \text{o.w.} \end{cases}$$

It is easy to see that $f_{\mathbf{x}}^\epsilon$ is $1-$Lipschitz. We consider three different cases,

1. If $\mathbf{z}_1, \mathbf{z}_2 \in B(\mathbf{x}, 2\epsilon)^c$ then $|f_{\mathbf{x}}^\epsilon(\mathbf{z}_1) - f_{\mathbf{x}}^\epsilon(\mathbf{z}_2)| = 0 \leq \|\mathbf{z}_1 - \mathbf{z}_2\|$. The result follows.

2. If $\mathbf{z}_1 \in B(\mathbf{x}, 2\epsilon)$ but $\mathbf{z}_2 \in B(\mathbf{x}, 2\epsilon)^c$. Let $z_3$ be the intersection point in the line going from $\mathbf{z}_1$ to $\mathbf{z}_2$ lying on $S(\mathbf{x}, 2\epsilon)$. Then $|f_{\mathbf{x}}^\epsilon(\mathbf{z}_1) - f_{\mathbf{x}}^\epsilon(\mathbf{z}_2)| = \min_{\mathbf{z}' \in S(\mathbf{x}, 2\epsilon)} \|\mathbf{z}_1 - \mathbf{z}'\|_2 \leq \|\mathbf{z}_1 - \mathbf{z}_3\|_2 \leq \|\mathbf{z}_1 - \mathbf{z}_2\|_2$.

3. If $\mathbf{z}_1, \mathbf{z}_2 \in B(\mathbf{x}, 2\epsilon)$. It is easy to see that $|f_{\mathbf{x}}^\epsilon(\mathbf{z}_1) - f_{\mathbf{x}}^\epsilon(\mathbf{z}_2)| = |\|\mathbf{z}_1 - \mathbf{x}\|_2 - \|\mathbf{z}_2 - \mathbf{x}\|_2|$. And therefore by the triangle inequality applied to $\mathbf{x}, \mathbf{z}_1, \mathbf{z}_2$, that $|\|\mathbf{z}_1 - \mathbf{x}\|_2 - \|\mathbf{z}_2 - \mathbf{x}\|_2| \leq \|\mathbf{z}_1 - \mathbf{z}_2\|_2$. The result follows.

Let $\mathcal{N}(B(\mathbf{0}, 1), 2\epsilon)$ be a $2\epsilon-$packing of the unit ball. For simplicity we'll use the notation $N_{2\epsilon} = |\mathcal{N}(B(\mathbf{0}, 1), 2\epsilon)|$. Define the set of functions $\mathcal{F}^\epsilon = \{f_{\mathbf{x}}^\epsilon \text{ for all } \mathbf{x} \in \mathcal{N}(B(\mathbf{0}, 1), 2\epsilon)\}$ and define $\mathcal{D}_{\mathcal{F}}$ as the uniform distribution over $\mathcal{F}^\epsilon \subset \mathcal{F}$. Similar to the case when $\mathcal{F}$ is the set of linear functions, we make use of Yao's principle. Let $\mathrm{DetAlg}$ be an optimal deterministic algorithm for $\mathcal{D}_{\mathcal{F}}$.

Let $a_i$ be $\mathrm{DetAlg}$'s $i-$th query point and $r_i$ be the $i-$th reward it receives. If the ground truth was $f_{\mathbf{x}}^\epsilon$ and the algorithm does not sample a query point from inside $B(\mathbf{x}, 2\epsilon)$, it will receive a reward of $0$ and thus would not have found an $\epsilon-$optimal point. Thus $t_{\mathrm{opt}}(f_{\mathbf{x}}^\epsilon) \geq$ first time to pull an arm in $B(\mathbf{x}, 1)$. As a consequence of this fact,

$$\mathbb{E}_{f_{\mathbf{x}}^\epsilon \sim \mathcal{D}_{\mathcal{F}}} \left[ \mathbf{1}(a_1 \in B(x, 2\epsilon)) \right] \leq \frac{1}{N_{2\epsilon}}.$$

Hence $\mathbb{E}_{f_{\mathbf{x}}^\epsilon \sim \mathcal{D}_{\mathcal{F}}} \left[ \mathbf{1}(a_1 \notin B(x, 2\epsilon)) \right] \geq \frac{N_{2\epsilon} - 1}{N_{2\epsilon}}$. Therefore,

$$\mathbb{E}\left[ t_{\mathrm{opt}}^\epsilon \right] \geq \sum_{\ell=1}^{N_{2\epsilon}} \frac{N_{2\epsilon} - \ell}{N_{2\epsilon} - \ell + 1}$$

$$\geq \sum_{\ell=1}^{N_{2\epsilon}/2} \frac{N_{2\epsilon} - \ell}{N_{2\epsilon} - \ell + 1}$$

$$\geq \frac{1}{4} N_{2\epsilon}.$$

Since $N_{2\epsilon} \overset{(i)}{\geq} \mathrm{Covering}(B(\mathbf{0}, 1), 4\epsilon) \overset{(ii)}{\geq} \left(\frac{1}{4\epsilon}\right)^d$ where inequality $(i)$ is a consequence of Lemma 5.5 and inequality $(ii)$ from Lema 5.7 in Wainwright (2019).

$\square$

**Translating to Quantile Optimality** . The results of Lemma D.1 can be interpreted in the langauge of quantile optimality by imposing a uniform measure over the sphere. In this case $\epsilon-$optimality is equivalent (approximately) to a $1 - 2^{-d(1-\epsilon)^2}$ quantile.

The results of Lemma D.1 hold regardless of the batch size $B$. It is thus impossible to design an algorithm that can single out an $\epsilon$-optimal arm in less than $\mathbb{T}^\epsilon(\mathcal{A}, \mathcal{F})$ queries for all problems defined by the pair $\mathcal{A}, \mathcal{F}$ simultaneously.

## D.2  OPTIMISM AND ITS PROPERTIES

The objective of this section is to prove Theorem 4.1 which we restart for the reader's convenience.

**Theorem 4.1.** *The regret of* OAE *with* $\mathbb{A}_{\max,b}$ *acquisition function satisfies,*

$$\mathrm{Reg}_{\mathcal{F},b}(T) = \mathcal{O}\left(\dim_E(\mathcal{F}, \alpha_T)b + \omega\sqrt{\dim_E(\mathcal{F}, \alpha_T)Tb}\right).$$

*With* $\alpha_t = \max\left(\frac{1}{t^2}, \inf\{\|f_1 - f_2\|_\infty : f_1, f_2 \in \mathcal{F}, f_1 \neq f_2\}\right)$ *and* $C = \max_{f \in \mathcal{F}, a, a' \in \mathcal{A}} |f(a) - f(a')|$.

Let's start by defining the $\epsilon-$Eluder dimension, a complexity measure introduced by Russo & Van Roy (2013) to analyze optimistic algorithms. Throughout this section we'll use the notation $\|g\|_A = \sqrt{\sum_{a \in A} g^2(a)}$ to denote the data norm of function $g : \mathcal{A} \to \mathbb{R}$.

**Definition D.2.** Let $\epsilon \geq 0$ and $\mathcal{Z} = \{a_i\}_{i=1}^n \subset \mathcal{A}$ be a sequence of arms.

1. An action $a$ is $\epsilon-$dependent on $\mathcal{Z}$ with respect to $\mathcal{F}$ if any $f, f' \in \mathcal{F}$ satisfying $\sqrt{\sum_{i=1}^n (f(a_i) - f'(a_i))^2} \leq \epsilon$ also satisfies $|f(a) - f'(a)| \leq \epsilon$.

2. An action $a$ is $\epsilon-$independent of $\mathcal{Z}$ with respect to $\mathcal{F}$ if $a$ is not $\epsilon-$dependent on $\mathcal{Z}$.

3. The $\epsilon-$eluder dimension $\dim_E(\mathcal{F}, \epsilon)$ of a function class $\mathcal{F}$ is the length of the longest sequence of elements in $\mathcal{A}$ such that for some $\epsilon' \geq \epsilon$, every element is $\epsilon'-$independent of its predecessors.

**Lemma D.3.** *Let's assume* $y_\star$ *satisfies* $\min_{f \in \mathcal{F}} \|y_\star - f\|_\infty \leq \omega$ *where* $\|g\|_\infty = \max_{a \in \mathcal{Z}} |g(a)|$. *Let* $\widetilde{y}_\star = \arg\min_{f \in \mathcal{F}} \|y_\star - f\|_\infty$ *be the* $\|\cdot\|_\infty$ *projection of* $y_\star$ *onto* $\mathcal{F}$. *If* $\widetilde{f}_t$ *is computed by solving 1 with* $\gamma_t \geq (t-1)\omega^2 b$ *with acquisition objective* $\mathbb{A}_{\max,b}$ *the response predictions of* $\widetilde{f}_t$ *over values* $B_t$ *satisfy,*

$$\sum_{a \in B_{\star,t}} y_\star(a) \leq b\omega + \sum_{a \in B_{\star,t}} \widetilde{y}_\star(a) \leq b\omega + \sum_{a \in B_t} \widetilde{f}_t(a).$$

*where* $B_{\star,t} = \max_{\mathcal{B} \subset \mathcal{U}_t, |\mathcal{B}|=b} \sum_{a \in \mathcal{B}} y_\star(a)$.

*Proof.* Let $\mathcal{F}_t$ be the subset of $\mathcal{F}$ satisfying $f \in \mathcal{F}_t$ if $\sum_{(x,y) \in \mathcal{D}_t} (f(x) - y)^2 \leq (t-1)\omega^2 \leq \gamma_t$. By definition $\widetilde{y}_\star \in \mathcal{F}_t$. Substituting the definition of $\mathbb{A}_{\max,b}$ into Equation 1,

$$\widetilde{f}_t \in \arg\max_{f \in \mathcal{F}_t} \max_{\mathcal{B} \subset \mathcal{U}_t, |\mathcal{B}|=b} \sum_{a \in \mathcal{B}} f(a).$$

Since $\widetilde{y}_\star \in \mathcal{F}_t$,

$$\max_{f \in \mathcal{F}_t, \mathcal{B} \subset \mathcal{U}_t, |\mathcal{B}|=b} \sum_{a \in \mathcal{B}} f(b) = \sum_{a \in B_t} \widetilde{f}_t(a) \geq \sum_{a \in \widetilde{B}_{\star,t}} \widetilde{y}_\star(a) \geq \sum_{a \in B_{\star,t}} \widetilde{y}_\star(a).$$

Where $\widetilde{B}_{\star,t} = \max_{\mathcal{B} \subset \mathcal{U}_t, |\mathcal{B}|=b} \sum_{a \in \mathcal{B}} \widetilde{y}_\star(a)$ and $B_{\star,t} = \max_{\mathcal{B} \subset \mathcal{U}_t, |\mathcal{B}|=b} \sum_{a \in \mathcal{B}} y_\star(a)$. Finally, for all $a \in \mathcal{A}$, we have $\widetilde{y}_\star(a) + \omega \geq y_\star(a)$. This finalizes the proof. m $\qquad\square$

Since $\widetilde{y}_\star(a) + \omega \geq y_\star(a)$. for all $a \in \mathcal{A}$, Lemma D.3 implies,

$$\sum_{\ell=1}^t \left(\sum_{a \in B_{\star,t}} y_\star(a) - \sum_{a \in B_t} y_\star(a)\right) \leq 2bt\omega + \sum_{\ell=1}^t \sum_{a \in B_t} \widetilde{f}_t(a) - \widetilde{y}_\star(a) \tag{13}$$

We define the width of a subset $\widetilde{\mathcal{F}} \subseteq \mathcal{F}$ at an action $a \in \mathcal{A}$ by,

$$r_{\widetilde{\mathcal{F}}}(a) = \sup_{f,f' \in \widetilde{\mathcal{F}}} |f(a) - f'(a)|.$$

And use the shorthand notation $r_t(a)$ to denote $r_{\mathcal{F}_t}(a)$ where $\mathcal{F}_t = \{f \in \mathcal{F} \text{ s.t. } \sum_{(a,y) \in \mathcal{D}_t} (f(a) - y)^2 \leq \gamma_t\}$. Equation 13 implies,

$$\sum_{\ell=1}^{t} \left( \sum_{a \in B_{\star,t}} y_\star(a) - \sum_{a \in B_t} y_\star(a) \right) \leq 2bt\omega + \sum_{\ell=1}^{t} \sum_{a \in B_t} r_t(a) \tag{14}$$

In order to bound the contribution of the sum $\sum_{\ell=1}^{t} \sum_{a \in B_t} r_t(a)$ we use a similar technique as in Russo & Van Roy (2013). First we prove a generalization of Proposition 3 of Russo & Van Roy (2013) to the case of parallel feedback.

**Proposition D.4.** *If $\{\gamma_t \geq 0 | t \in \mathbb{N}\}$ is a nondecreasing sequence and $\mathcal{F}_t = \{f \in \mathcal{F} : \sum_{(a,y) \in \mathcal{D}_t} (f(a) - y)^2 \leq \gamma_t\}$ then,*

$$\sum_{t=1}^{T} \sum_{a \in B_t} \mathbf{1}(r_t(a) > \epsilon) \leq \mathcal{O}\left( \frac{\gamma_t d}{\epsilon^2} + bd \right).$$

*Where $d = \dim_E(\mathcal{F}, \epsilon)$.*

*Proof.* We will start by upper bounding the number of disjoint sequences in $\mathcal{D}_{t-1}$ that an action $a \in B_t$ can be $\epsilon-$dependent on when $r_t(a) > \epsilon$.

If $r_t(a) > \epsilon$ there exist $\bar{f}, \underline{f} \in \mathcal{F}_t$ such that $\bar{f}(a) - \underline{f}(a) > \epsilon$. By definition if $a \in B_t$ is $\epsilon-$dependent on a sequence $S \subseteq \cup_{\ell=1}^{t-1} B_\ell = \mathcal{D}_t$ then $\|\bar{f} - \underline{f}\|_S^2 > \epsilon^2$ (since otherwise $\|\bar{f} - \underline{f}\|_S^2 \leq \epsilon^2$ would imply $a$ to be $\epsilon-$independent of $\mathcal{D}_t$). Thus if $a$ is $\epsilon-$dependent on $K$ disjoint sequences $S_1, \cdots, S_K \subseteq \mathcal{D}_t$, then $\|\bar{f} - \underline{f}\|_{\mathcal{D}_t}^2 \geq K\epsilon^2$. By the triangle inequality,

$$\|\bar{f} - \underline{f}\|_{\mathcal{D}_t} \leq \|\bar{f} - \widetilde{y}_\star\|_{\mathcal{D}_t} + \|\underline{f} - \widetilde{y}_\star\|_{\mathcal{D}_t} \leq 2\sqrt{\gamma_t}.$$

Thus it follows that $\epsilon\sqrt{K} < 2\sqrt{\gamma_t}$ and therefore

$$K \leq \frac{4\gamma_t}{\epsilon^2} \tag{15}$$

Next we prove a lower bound for $K$. In order to do so we prove a slightly more general statement. Consider a batched sequence of arms $\{\bar{a}_{\ell,i}\}_{i \in [b], \ell \in [\tau]}$ where for the sake of the argument $\bar{a}_{\ell,i}$ is not necessarily meant to be $a_{\ell,i}$. We use the notation $\bar{B}_\ell = \{\bar{a}_{\ell,i}\}_{i \in [b]}$ to denote the $\ell-$th batch in $\{\bar{a}_{\ell,i}\}_{i \in [b], \ell \in [\tau]}$ and $\bar{\mathcal{D}}_t = \cup_{\ell=1}^{t-1} \bar{B}_\ell$.

Let $\tau \in \mathbb{N}$ and define $\widetilde{K}$ as the largest integer such that $\widetilde{K}d + b \leq \tau b$. We show there is a batch number $\ell' \leq \tau$ and in-batch index $i'$ such that $a_{\ell',i'}$ is $\epsilon-$dependent on a subset of disjoint sequences of size at least $\frac{\widetilde{K}}{2}$ out of a set of $\widetilde{K}$ disjoint sequences $\bar{S}_1, \cdots, \bar{S}_{\widetilde{K}} \subseteq \bar{\mathcal{D}}_{\ell-1}$.

First let's start building the $\bar{S}_i$ sequences by setting $\bar{S}_i =$ the $i-$th element in $\{\bar{a}_{\ell,i}\}_{i \in [b], \ell \in [\tau]}$ ordered lexicographically. This will involve elements of up to batch $\bar{B}_{\lceil \widetilde{K}/b \rceil}$.

Since we are going to apply the same argument recursively in our proof, let's denote the 'current' batch index in the construction of $\bar{S}_1, \cdots, \bar{S}_{\widetilde{K}}$ as $\bar{\ell}$, this is, we set $\bar{S}_1, \cdots, \bar{S}_{\widetilde{K}} \subseteq \bar{\mathcal{D}}_{\bar{\ell}}$. At the start of the argument $\bar{\ell} = \lceil \widetilde{K}/b \rceil$.

If there is an arm $a \in \bar{B}_{\bar{\ell}+1}$ such that $a$ is $\epsilon-$dependent on at least $\widetilde{K}/2$ of the $\{\bar{S}_i\}_{i=1}^{\widetilde{K}}$ sequences, the result would follow. Otherwise, it must be the case that for all $a \in \bar{B}_{\bar{\ell}+1}$ there are at least $\frac{\widetilde{K}}{2}$ sequences in $\{\bar{S}_i\}_{i=1}^{\widetilde{K}}$ on which $a$ is $\epsilon-$independent.

Let's consider a bipartite graph with edge sets $\bar{B}_{\bar{\ell}+1}$ and $\{\bar{S}_i\}_{i=1}^{\widetilde{K}}$. We draw an edge between $a \in \bar{B}_{\bar{\ell}+1}$ and $\bar{S}_j \in \{\bar{S}_i\}_{i=1}^{\widetilde{K}}$ if $a$ is $\epsilon$−independent on $\bar{S}_j$. If for all $a \in \bar{B}_{\bar{\ell}+1}$ there are at least $\frac{\widetilde{K}}{2}$ sequences in $\{\bar{S}_i\}_{i=1}^{\widetilde{K}}$ on which $a$ is $\epsilon$−independent, Lemma D.5 implies the existence of a matching of size at least $\min(\frac{\widetilde{K}}{8}, b)$ between elements in $\bar{B}_{\bar{\ell}+1}$ and the sequences in $\{\bar{S}_i\}_{i=1}^{\widetilde{K}}$.

The case $\min(\frac{\widetilde{K}}{8}, b) = \frac{\widetilde{K}}{8}$ can only occur when $\frac{(\tau-1)b}{8d} \leq b$ and therefore when $\tau b \leq 8bd + b$.

In case $\min(\frac{\widetilde{K}}{8}, b) = b$. If we reach $\bar{\ell} = \tau$ it must be the case that at least $(\tau - 1)b$ points could be accommodated into the $\widetilde{K}$ sequences. By definition of $\widetilde{K}$ it must be the case that each sub-sequence $S_i$ satisfies $|S_i| \geq d$. Since each element of subsequence $S_i$ is $\epsilon$−independent of its predecesors, $|S_i| = d$. In this case we would conclude there is an element in $\bar{B}_\tau$ that is $\epsilon$−dependent on $\widetilde{K}$ and therefore at least $\frac{\widetilde{K}}{2} = \frac{(\tau-1)b}{2d}$ subsequences. If $\tau b \geq \frac{8\gamma_\tau d}{\epsilon^2} + 2d + b$ then $\frac{\widetilde{K}}{2} \geq \frac{4\gamma_\tau}{\epsilon^2} + 1$.

Combining the results of the two last paragraphs we conclude that if $\tau b \geq \frac{8\gamma_\tau d}{\epsilon^2} + 2d + 2b + 8bd$ then there is a batch index $\ell' \in [\tau]$ such that there is an arm $\bar{a}_{\ell',i} \in \bar{B}_{\ell'}$ that is $\epsilon$−dependent of at least $\frac{\widetilde{K}}{2} \geq \frac{4\gamma_\tau}{\epsilon^2} + 1$ disjoint sequences contained in $\bar{\mathcal{D}}_{\ell'}$.

Let's apply the previous result to the sequence of $a_{t,i}$ for $i \in [b]$ and $t \in \mathbb{R}$ such that $r_t(a_{t,i}) > \epsilon$. An immediate consequence of the previous results is that if $\sum_{\ell=1}^t \sum_{a \in B_\ell} \mathbf{1}(r_\ell(a) > \epsilon) \geq \frac{8\gamma_t d}{\epsilon^2} + 2d + 2b + 8bd$ there must exist an arm in $B_t$ such that it is $\epsilon$−dependent on at least $\frac{4\gamma_t}{\epsilon^2} + 1$ disjoint sequences of $\mathcal{D}_t$. Nonetheless, Equation 15 implies this is impossible. Thus, $\sum_{\ell=1}^t \sum_{a \in B_\ell} \mathbf{1}(r_\ell(a) > \epsilon) \leq \frac{8\gamma_t d}{\epsilon^2} + 2d + 2b + 8bd \leq \frac{8\gamma_t d}{\epsilon^2} + 12bd$. The result follows. $\qquad\square$

Finally, the RHS of equation 14 can be upper bounded using a modified version of Lemma 2 of Russo & Van Roy (2013) (which can be found in Appendix D.2 ) yielding,

$$\sum_{\ell=1}^T \left( \sum_{a \in B_{\star,t}} y_\star(a) - \sum_{a \in B_t} y_\star(a) \right) \leq \frac{1}{T} + \mathcal{O}\left( \min\left(\dim_E(\mathcal{F}, \alpha_T)b, T\right)C + \sqrt{\dim_E(\mathcal{F}, \alpha_T)}\omega T \right).$$

Where $\alpha_t = \max\left( \frac{1}{t^2}, \inf\{\|f_1 - f_2\|_\infty : f_1, f_2 \in \mathcal{F}, f_1 \neq f_2\} \right)$ and $C = \max_{f \in \mathcal{F}, a, a' \in \mathcal{A}} |f(a) - f(a')|$. The quantity in the left is known as regret. This result implies the regret is bounded by a quantity that grows linearly with $\omega$, the amount of misspecification but otherwise only with the scale of $\dim_E(\mathcal{F}, \alpha_T)b$. Our result is not equivalent to splitting the datapoints in $b$ parts and adding $b$ independent upper bounds. The resulting upper bound in the later case will have in a term of the form $\mathcal{O}\left( b\sqrt{\dim_E(\mathcal{F}, \alpha_T)}\omega T \right)$ whereas in our analysis this term does not depend on $b$. When $\omega = 0$, the regret is upper bounded by $\mathcal{O}\left( \dim_E(\mathcal{F}, \alpha_T)d \right)$. For example, in the case of linear models, the authors of Russo & Van Roy (2013) show $\dim_E(\mathcal{F}, \alpha_T) = \mathcal{O}(d \log(1/\epsilon))$. This shows that for example sequential OAE when $b = 1$ achieves the lower bound of Lemma D.1 up to logarithmic factors. In the setting of linear models, the $b$ dependence in the rate above is unimprovable by vanilla OAE without diversity aware sample selection. This is because an optimistic algorithm may choose to use all samples in each batch to explore explore a single unexplored one dimensional direction. Theoretical analysis for $\mathrm{OAE} - \mathrm{DvD}$ and $\mathrm{OAE} - \mathrm{Seq}$ is left for future work.

**Lemma D.5.** *Let $\mathcal{G}$ be a bipartite graph with node set $A \cup B$ such that $|A| = \tilde{K}$ and $|B| = b$. If for all nodes $v \in B$ it follows that $N(v) \geq \frac{\widetilde{K}}{2}$ then,*

1. *If $\frac{\widetilde{K}}{2} \geq b$ there exists a perfect matching between the nodes in $B$ and the ones in $A$.*

2. *If instead $\frac{\widetilde{K}}{2} < b$ there exists a subset of $\frac{\widetilde{K}}{8}$ nodes in $A$ with a perfect matching to $B$.*

*Proof.* The first item follows immediately from Hall's marriage theorem. Notice that in this case the neighboring set of any subset of nodes in $B$ has a cardinality of at least $\frac{\widetilde{K}}{2}$ and therefore it is at least the size of $B$. The conditions of Hall's theorem are satisfied thus implying the existence of a perfect matching between the nodes in $B$ to the nodes in $A$.

In the second scenario, let's first prove there exists a subset $A'$ of $A$ of size at least $\frac{\widetilde{K}}{4}$ such that every element in $A'$ has at least $\frac{b}{4}$ neighbors in $B$. We prove this condition by the way of contradiction. There are at least $\frac{b\widetilde{K}}{2}$ edges in the graph. Suppose there were at most $L$ vertices in $A$ with degree greater or equal to $\frac{b}{4}$. This value of $L$ must satisfy the inequality,

$$Lb + (\widetilde{K} - L)\left(\frac{b}{4} - 1\right) \geq \frac{b\widetilde{K}}{2}.$$

This is because the maximum number of edges a vertex in $A$ can have equals $b$. Thus, $L \geq \frac{\widetilde{K}}{3} - 1$.

All nodes in $L$ have degree at least $\frac{b}{4}$. If we restrict ourselves to a subset $\widetilde{L}$ of $L$ of size $\frac{\widetilde{K}}{8}$ and since in this scenario $\frac{\widetilde{K}}{8} < \frac{b}{4}$, Hall's stable marriage theorem implies there is a perfect matching between $\widetilde{L}$ to $B$. The result follows.

$\square$

Lemma 2 of Russo & Van Roy (2013) adapted to notation of OAE.

**Lemma D.6.** *If $\{\gamma_t \geq 0 | t \in \mathbb{N}\}$ is a nondecreasing sequence and $\mathcal{F}_t = \{f \in \mathcal{F} \text{ s.t. } \sum_{(a,y)\in\mathcal{D}_t}(f(a) - y)^2 \leq \gamma_t\}$ then with probability 1 for all $T \in \mathbb{N}$,*

$$\sum_{t=1}^{T} \sum_{a \in B_t} r_t(a) \leq \mathcal{O}\left(\dim_E(\mathcal{F}, \alpha_T)b + \omega\sqrt{\dim_E(\mathcal{F}, \alpha_T)Tb}\right).$$

*Where $\alpha_t = \max\left(\frac{1}{tb}, \frac{1}{2}\inf\{\|f_1 - f_2\|_\infty : f_1, f_2 \in \mathcal{F}, f_1 \neq f_2\}\right)$ and $C = \max_{f\in\mathcal{F}, a, a'\in\mathcal{A}}|f(a) - f(a')|$.*

*Proof.* The proof of lemma D.6 follows the proof template as that of Lemma 2 in Russo & Van Roy (2013). We reproduce it here for completeness.

For ease of notation we use $d = \dim_E(\mathcal{F}, \alpha_T)$. We first re-order the sequence $\{r_1(a_{1,1}), r_1(a_{1,2}), \cdots, r_1(a_{1,b}), \cdots, r_T(a_{T,b}))$ as $r_{i_1} \leq \cdots \leq r_{i_{Tb}}$. We have,

$$\sum_{t=1}^{T}\sum_{a\in B_t} r_t(a) = \sum_{j=1}^{Tb} r_{i_j} = \sum_{j=1}^{Tb}\mathbf{1}(r_{i_j} \leq \alpha_T)r_{i_j} + \sum_{j=1}^{Tb}\mathbf{1}(r_{i_j} > \alpha_T)r_{i_j} \overset{(i)}{\leq} 1 + \sum_{j=1}^{Tb}\mathbf{1}(r_{i_j} > \alpha_T)r_{i_j}$$

Where inequality $(i)$ holds by definition of $\alpha_T$ noting that

$$\sum_{j=1}^{Tb}\mathbf{1}(r_{i_j} \leq \alpha_T)r_{i_j} = 0, \quad \text{if } \alpha_T = \frac{1}{2}\inf\{\|f_1 - f_2\|_\infty : f_1, f_2 \in \mathcal{F}, f_1 \neq f_2\}.$$

and

$$\sum_{j=1}^{Tb}\mathbf{1}(r_{i_j} \leq \alpha_T)r_{i_j} \leq 1, \quad \text{if } \alpha_T = \frac{1}{Tb}.$$

Proposition D.4 (since $d \geq \dim_E(\mathcal{F}, \epsilon)$ for all $\epsilon \geq \alpha_T$) implies that for all $i_j$ with $r_{i_j} > \alpha_T$, we can bound $j$ as

$$j \leq \sum_{t=1}^{T}\sum_{a\in B_t}\mathbf{1}(r_t(a) > r_{i_j}) \leq \mathcal{O}\left(\frac{\gamma_T d}{r_{i_j}^2} + bd\right)$$

---

**Algorithm 4** Noisy Batch Selection Principle (OAE)

---

**Input** Action set $\mathcal{A} \subset \mathbb{R}^d$, num batches $N$, batch size $b$
**Initialize** Observed points and labels dataset $\mathcal{D}_1 = \emptyset$
**for** $t = 1, \cdots, N$ **do**
    **If** $t = 1$ **then:**
    · Sample uniformly a size $b$ batch $B_t \sim \mathcal{U}_1$.
    **Else:**
    · Solve for $\widetilde{f}_t$ and compute $B_t \in \arg\max_{\mathcal{B} \subset \mathcal{U}_t || \mathcal{B}| = b} \mathbb{A}(\widetilde{f}_t, \mathcal{B})$.
    Observe batch rewards $Y_t = \{y_*(a) \text{ for } a \in B_t\}$
    Update $\mathcal{D}_{t+1} = \mathcal{D}_t \cup \{(B_t, Y_t)\}$ and $\mathcal{U}_{t+1} = \mathcal{U}_t \backslash B_t$ .

---

So there is a constant $c > 0$ such that $r_{i_j} \leq \mathcal{O}\left(\sqrt{\frac{\gamma_T d}{(j - cbd)_+}}\right)$. For all $j \leq cbd$ notice that $\sum_{j=1}^{cbd} r_{i_j} = \mathcal{O}(bd)$ since the radii $r_{i_j}$ are all of constant size. We conclude that,

$$\sum_{j=1}^{Tb} r_{i_j} \leq \mathcal{O}\left(bd + \sum_{j=cbd+1}^{Tb} \sqrt{\frac{\gamma_T d}{(j - cbd)_+}}\right)$$

$$\leq \mathcal{O}\left(bd + \sqrt{\gamma_T d} \int_{j=1}^{Tb - cbd} \sqrt{k} dk\right)$$

$$= \mathcal{O}\left(bd + \sqrt{\gamma_T d Tb}\right).$$

Substituting $\gamma_T = \omega^2 Tb$ we conclude,

$$\sum_{j=1}^{Tb} r_{i_j} = \mathcal{O}\left(bd + \omega \sqrt{d} Tb\right).$$

Thus finalizing the result. $\qquad\square$

Combining the result of Lemma D.6 and Equation 14 finalizes the proof of Theorem 4.1.

### D.3 NOISY RESPONSES

In this section we describe how our results imply improved bounds for the setting with noisy responses. In this case we assume that $y_{t,i} = y_\star(a_{t,i}) + \xi_{t,i}$ where $\xi_{t,i}$ is conditionally zero mean.

We consider the following optimistic least squares algorithm,

## E EXPERIMENTS

We demonstrate the effectiveness of our OAE algorithm in several problem settings across public and synthetic datasets. We evaluate the algorithmic implementations described in Section 4.1 by setting the acquisition function to $\mathbb{A}_{\mathrm{avg}}(f, \mathcal{U})$ and the batch selection rule as in Equation 2 for the vanilla OAE methods and as in Equations 6 and 8 for OAE's diversity inducing versions $\mathrm{OAE} - \mathrm{DvD}$ and $\mathrm{OAE} - \mathrm{Seq}$ respectively. All neural network architectures use ReLU activations. All ensemble methods use an ensemble size of $M = 10$ and xavier parameter initialization.

**Small vs Large Batch Regimes.** Oftentimes the large batch - small number of iterations regime is the most interesting scenario from a practical perspective Hanna & Doench (2020). In scientific settings like pooled genetic perturbations, each experiment may take a long time (many weeks or months) to conclude, but it is possible to conduct a batch of experiments in parallel together. We study this regime in Section 5.

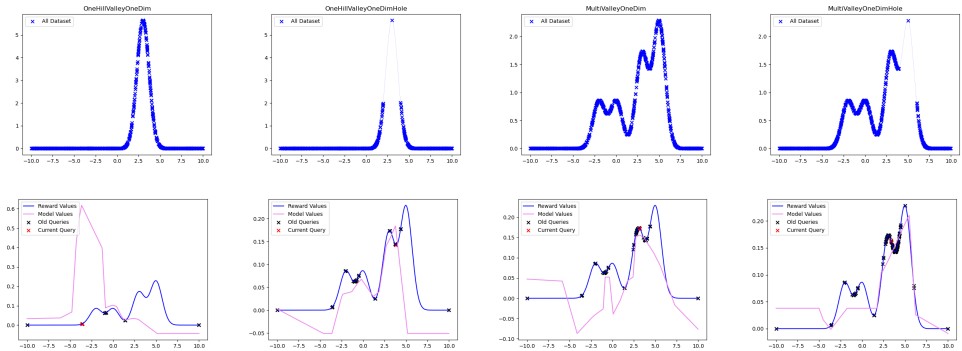

Figure 7: (top row) Synthetic one dimensional datasets (bottom) Evolution of $\mathrm{MeanOpt}$ in the MultiValleyOneDimHole dataset using $\lambda_{\mathrm{reg}} = 0.001$. *From left to right:* Iterations $5, 15, 25, 45$.

### E.1 VANILLA OAE

We test $\mathrm{MeanOpt}$, $\mathrm{HingePNorm}$, $\mathrm{MaxOpt}$, $\mathrm{Ensemble}$ and $\mathrm{EnsembleNoiseY}$'s performance (see Section 4.1 for a detailed description of each of these algorithms) over different values of the regularization parameter $\lambda_{\mathrm{reg}}$, including the 'greedy' choice of $\lambda_{\mathrm{reg}} = 0$, henceforth referred to as Greedy. We compare these algorithms to the baseline method that selects points $B_t$ by selecting a uniformly random batch of size $B$ from the set $\mathcal{U}_t$, henceforth referred to as $\mathrm{RandomBatch}$ and against each other.

We conduct experiments on three kind of datasets. First in Section E.1.1 we capture the behavior of OAE in a set of synthetic one dimensional datasets specifically designed to showcase different landscapes for $y_\star$ ranging from uni-modal to multi-modal with missing values. In Section E.1.2 we conduct similar experiments on public datasets from the UCI database Dua & Graff (2017). In both sections E.1.1 and E.1.2 all of our experiments have a batch size of 3, a time horizon of $N = 150$ and over two types of network architectures. In Section 5 we consider the setting in which we have observed the effect of knockouts in one biological context (i.e., cell line) and would like to use it to plan a series of knockout experiments in another. We test OAE in this context by showing the effectiveness of the $\mathrm{MeanOpt}$, $\mathrm{HingePNorm}$, $\mathrm{MaxOpt}$, $\mathrm{Ensemble}$ and $\mathrm{EnsembleNoiseY}$ methods in successfully leveraging the learned features from a source cell line in the optimization of a particular cellular proliferation phenotype for several target cell lines.

All of our vanilla OAE methods show that better expressivity of the underlying model class $\mathcal{F}$ allows for better performance (as measured by the number of trials it requires to find a response within a particular response quantile from the optimum). Low capacity (in our experiments ReLU neural networks with two layers of sizes of 10 and 5) models have a harder time competing against $\mathrm{RandomBatch}$ than larger ones (ReLU neural networks with two layers of sizes 100 and 10). We also present results for linear and 'very high' capacity models (two layer of sizes 300 and 100).

### E.1.1 SYNTHETIC ONE DIMENSIONAL DATASETS

Figure 7 shows different one dimensional synthetic datasets that used to validate our methods. The leftmost, the OneHillValleyOneDim dataset consists of 1000 arms uniformly sampled from the interval $[-10, 10]$. The responses $y$ are unimodal. The learner's goal is to find the arm with $x-$coordinate value equals to 3 as it is the one achieving the largest response. We use the dataset OneHillValleyOneDimHole to test for OAE's ability to find the maximum when the surrounding points are not present in the dataset.

The remaining two datasets MultiValleyOneDim and MultiValleyOneDimHole are built with the problem of multimodal optimization in mind. Each of these datasets have 4 local maxima. We use MultiValleyOneDim to test the OAE's ability to avoid getting stuck in local optima. The second dataset mimics the construction of the OneHillValleyOneDimHole dataset and on top of testing the algorithm's ability to escape local optima, it also is meant to test what happens when the global optimum's neighborhood isn't present in the dataset. Since one of the algorithms we test is the greedy algorithm (corresponding to $\lambda = 0$), the 'Hole' datasets are meant to present a challenging situation for this class of algorithms.

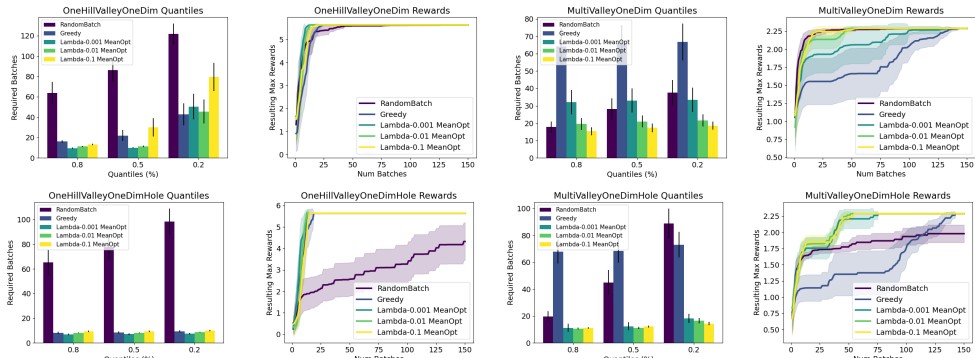

Figure 8: **NN 10-5**. $\tau$-quantile batch convergence (**left**) and corresponding rewards over batches (**right**) on the OneHillValleyOneDim (easier) and MultiValleyOneDim (harder), OneHillValleyOneDimHole and MultiValleyOneDimHole synthetic datasets show how the OAE algorithm can achieve higher reward faster than RandomBatch or Greedy. The neural network architecture has two hidden layers of sizes 10 and 5.

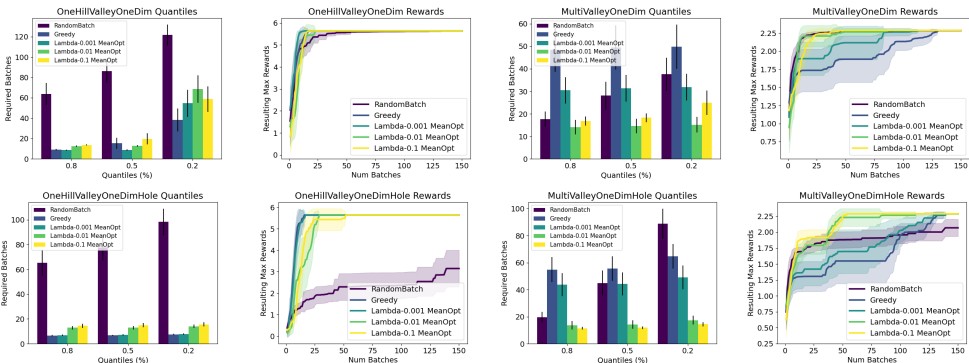

Figure 9: **NN 100-10**. $\tau$-quantile batch convergence (**left**) and corresponding rewards over batches (**right**) on the suite of synthetic datasets when the neural network architecture has two hidden layers of sizes 100 and 10.

We test several neural network architectures across all experiments. In all these cases we use ReLU activation functions trained for 5000 steps via the Adam optimizer using batches of size 10. In our tests we use a batch size $B = 3$, a number of batches $N = 150$ and repeat each experiment a total of 25 times, reporting average results with standard error bars at each time step.

First we compare the MeanOpt version of OAE with a simple RandomBatch strategy that selects a random batch of size $B$ from the dataset points that have not been queried yet and a Greedy algorithm corresponding to MeanOpt with $\lambda_{\mathrm{reg}} = 0$. Figures 8, 9 and 10 show representative results for MeanOpt with ($\lambda_{\mathrm{reg}} = 0.001, 0.01, 0.1$) accross three different neural network architectures, **NN 10-5**, **NN 100-10**, and **NN 300-100**. In all cases, the high optimism versions of MeanOpt perform substantially better than RandomBatch. In both multimodal datasets Greedy underperforms with respect to the versions of MeanOpt with $\lambda_{\mathrm{reg}} > 0$. This points to the usefulness of optimism when facing multimodal optimization landscapes. We also note that for example **NN 10-5** is the best performing architecture for MeanOpt with $\lambda_{\mathrm{reg}} = 0.001$ despite the regression loss of fitting the MultiValleyOneDim's responses with a **NN 10-5** architecture not reaching zero (see Figure 13). This indicates that the function class F need not contain $y_\star$ for MeanOpt to achieve good performance.

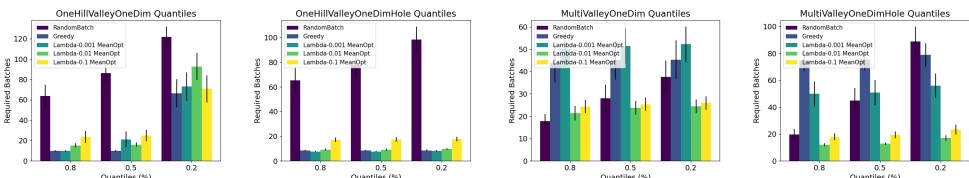

Figure 10: **NN 300-100**. $\tau$-quantile batch convergence on the suite of synthetic datasets when the network architecture has two hidden layers of sizes 300 and 100.

Moreover, it also indicates the use of higher capacity models, despite being able to achieve better regression fit may not perform better than lower capacity ones in the task of finding a good performing arm. We leave the task of designing smart strategies to select the optimal network architecture for future work. It suffices to note all of the architectures used in our experimental evaluation performed better than more naive strategies such as RandomBatch.

Second, in Figures 11 and 12 we evaluate MeanOpt vs ensemble implementations of OAE across the two neural network architectures **NN 10-5** and **NN 100-10**. We observe that Ensemble performs competitively with all other methods in the one dimensional datasets and outperforms all in the OneHillValleyOneDimHole. In the multi dimensional datasets, MeanOpt performs better than Ensemble, EnsembleNoiseY and Greedy. In this case the most optimistic version of MeanOpt ($\lambda_{\mathrm{reg}} = .1$) is the best performing of all. This may indicate that in multi modal environments the optimism injected by the random initialization of the ensemble models in Ensemble or the reward noise in EnsembleNoiseY do not induce an exploration strategy as effective as the explicit optimistic fit of MeanOpt. In unimodal datasets, the opposite is true, MeanOpt with a large regularizer ($\lambda_{\mathrm{reg}} = 0.1$) underperforms Ensemble, EnsembleNoiseY and Greedy. In Appendix G.1 the reader may find similar results for MaxOpt and HingeOpt. Similar results hold in that case.

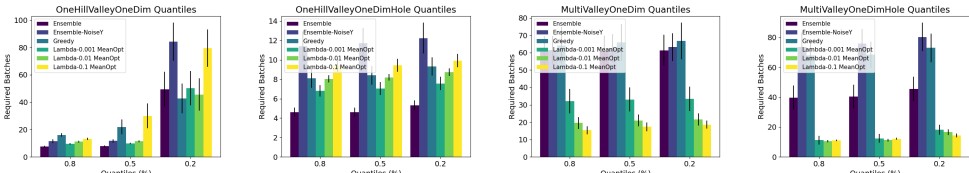

Figure 11: **NN 10-5**. $\tau$-quantile batch convergence of MeanOptimism vs EnsembleOptimism vs EnsembleNoiseY.

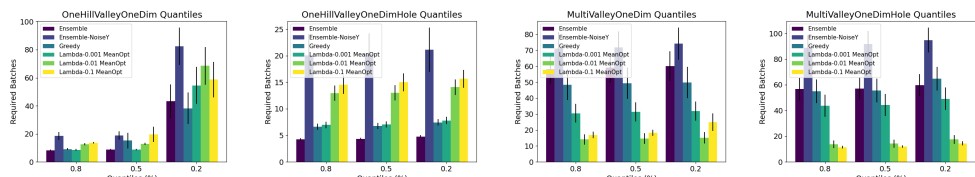

Figure 12: **NN 100-10**. $\tau$-quantile batch convergence of MeanOptimism vs EnsembleOptimism vs EnsembleNoiseY.

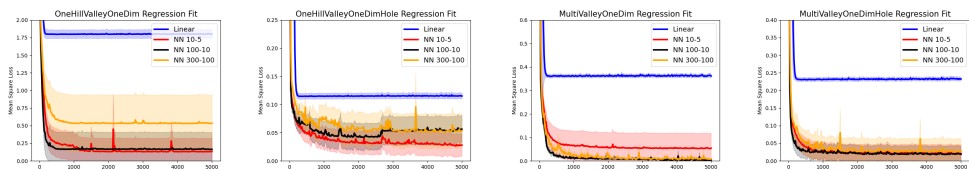

Figure 13: **Training set** regression fit curves evolution over training for the suite of synthetic datasets. Each training batch contains 10 datapoints.

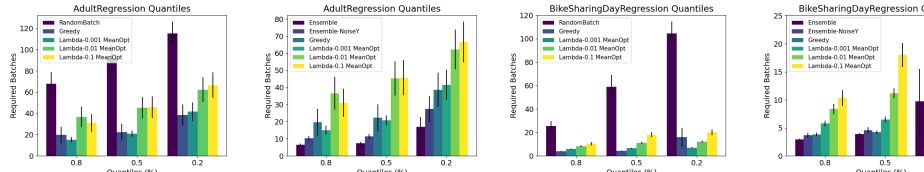

Figure 15: **NN 10-5**. $\tau$-quantile batch convergence on the Adult and BikeSharingDay datasets with regression-fitted response values.

### E.1.2 PUBLIC SUPERVISED LEARNING DATASETS

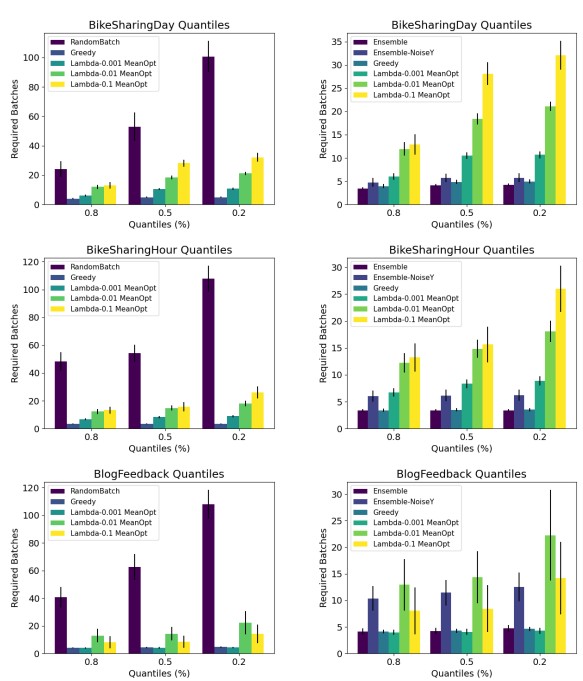

We test our methods on public binary classification (Adult, Bank) and regression (BikeSharingDay, BikeSharingHour, BlogFeedback) datasets from the UCI repository (Dua & Graff, 2017). In our implementation, the versions of the UCI datasets we use have the following characteristics. Due to our internal data processing that splits the data into train, test and validation sets the number of datapoints we consider may be different from the size of their public versions. Our code converts the datasets categorical attributes into numerical ones using one hot encodings. That explains the discrepancy between the number of attributes listed in the public description of these datasets and ours (see `https://archive.ics.uci.edu/ml/index.php`). The BikeSharingDay dataset consists of 658 datapoints each with 13 attributes. The BikesharingHour dataset consists of 15642 datapoints each with 13 attributes. The BlogFeedback dataset consists of 52396 datapoints each with 280 attributes. The Adult dataset consists of 32561 datapoints

Figure 14: **NN 100-10**. $\tau$-quantile batch convergence (**left**) BikeSharingDay, BikeSharingHour and BlogFeedback datasets with original response values.

each with 119 attributes. The Bank dataset (Moro et al., 2014) consists of 32950 datapoints each with 60 attributes. To evaluate our algorithm we assume the response (regression target or binary classification label) values are noiseless. We consider each observation $i$ in a dataset to represent a discrete action, each of which has features $a_i$ and reward $y_*(a_i)$ from the response. In all of our experiments we use a batch size of 3 and evaluate over 25 independent runs each with 150 batches.

We first use all 5 public datasets to test OAE in the setting when the true function class $\mathcal{F}$ is known (in this case, a neural network) is known contain the function OAE learns over the course of the batches. We train a neural network under a simple mean squared error regression fit to the binary responses (for the binary classification datasets) or real-valued responses (for the regression datasets). This regression neural network consists of a neural network with two hidden layers. In Figure 15 we present results where the neural network layers have sizes 10 and 5 and the responses are fit to the Adult classification dataset and the regression dataset BikeSharingDay. In Appendix G.2 and Figure 32 we present results where we fit a two layer neural network model of dimensions 100 and 10 to the responses of the Bank classification dataset and the BikeSharingHour regression dataset. In each case we train the regression fitted responses on the provided datasets using $5,000$ batch gradient

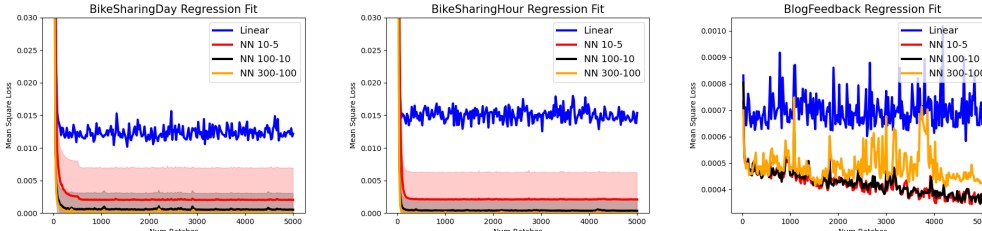

Figure 16: **Training set** regression fit curves evolution over training for the UCI datasets. Each training batch contains 10 datapoints.

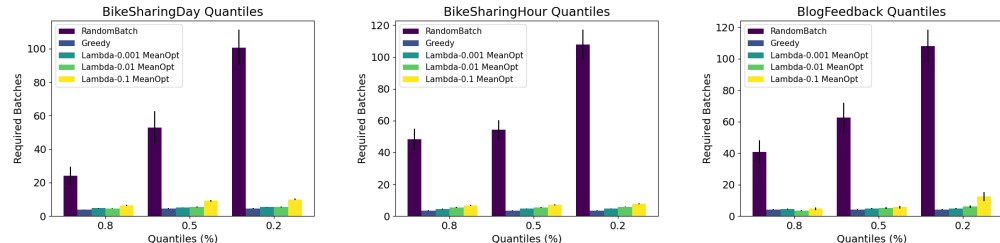

Figure 17: **Linear**. $\tau$-quantile batch convergence on the BikeSharingDay, BikeSharingHour and BlogFeedback datasets.

steps (with a batch size of 10). During test time we use the real-valued response predicted by our regression model as the reward for the corresponding action. This ensures the dataset's true reward response model has the same architecture as the reward model used by OAE.

We use the same experimental parameters and comparison algorithms as in the synthetic dataset experiments. Figure 15 shows the results on the binary classification $\mathrm{Adult}$ and regression $\mathrm{BikeSharingDay}$ datasets using these fitted responses. We observe the $\mathrm{MeanOpt}$ algorithm handily outperforms $\mathrm{RandomBatch}$ on both datasets. Appendix G.2 shows similar results for $\mathrm{BikeSharingHour}$ and $\mathrm{Bank}$. We also compare the performance of $\mathrm{MeanOpt}$, $\mathrm{Ensemble}$ and $\mathrm{EnsembleNoiseY}$ in the $\mathrm{Adult}$ and $\mathrm{BikeSharingDay}$ datasets. In both cases, ensemble methods achieved better performance than $\mathrm{MeanOpt}$. It remains an exciting open question to verify whether these observations translates into a general advantage for ensemble methods in the case when $y_\star \in \mathcal{F}$.

Given OAE's strong performance when the true and learned reward functions are members of the same function class $\mathcal{F}$, we next explore the performance when they are not necessarily in the same class by revising the problem on the regression datasets to use their original, real-world responses. Figure 14 shows results for the $\mathrm{BikeSharingDay}$, $\mathrm{BikeSharingHour}$ and $\mathrm{BlogFeedback}$ datasets. In this case $\mathrm{Ensemble}$ outperforms both $\mathrm{RandomBatch}$ in $\tau$-quantile convergence time. In all of these plots we observe that high optimism approaches underperform compared with low optimism ones. $\mathrm{Ensemble}$ and $\mathrm{Greedy}$ (the degenerate version $\lambda_{\mathrm{reg}} = 0$ of $\mathrm{MeanOpt}$) achieve the best performance across all three datasets. We observe the same phenomenon take place even when F is a class of linear functions (see figure 18). Just as we observed in the case of the suite of synthetic datasets, setting $\mathcal{F}$ to be a class of linear models $\mathrm{MeanOpt}$ still achieves substantial performance gains w.r.t $\mathrm{RandomBatch}$ (see figure 17) despite its regression fit loss never reaching absolute zero (see figure 16). In Appendix G.3, figure 33 we compare the performance of $\mathrm{MaxOpt}$ and $\mathrm{HingeOpt}$ with $\mathrm{RandomBatch}$ when F is a class of neural networks with hidden layers of sizes 100 and 10. We observe the performance of $\mathrm{MaxOpt}$, although beats $\mathrm{RandomBatch}$ is suboptimal in comparison with $\mathrm{MeanOpt}$. In contrast $\mathrm{HingeOpt}$ has a similar performance to $\mathrm{MeanOpt}$.

### E.2 EXPERIMENTS DIVERSITY SEEKING OBJECTIVES

In this section we explore how diversity inducing objectives can sometimes result in better performing variants of OAE. We implement and test $\mathrm{DetD}$, $\mathrm{SeqB}$, $\mathrm{Ensemble} - \mathrm{SeqB}$ and

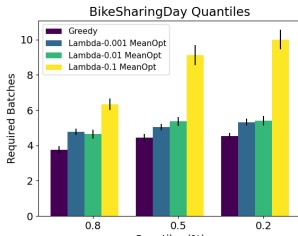 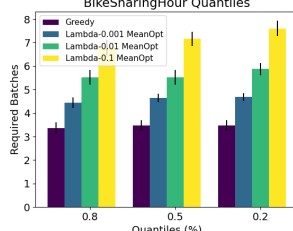 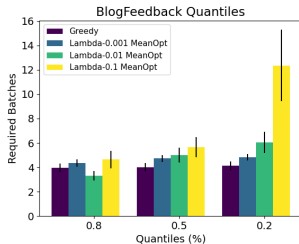

Figure 18: **Linear**. $\tau$-quantile batch convergence on the BikeSharingDay, BikeSharingHour and BlogFeedback datasets.

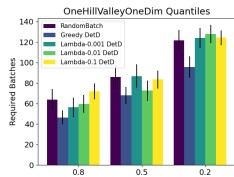 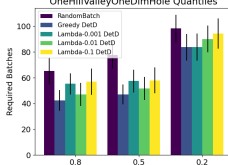 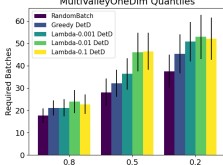 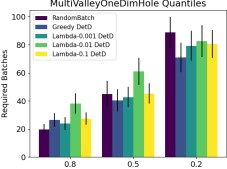

Figure 20: **NN 10-5**.$\tau$-quantile batch convergence performance of DetD vs RandomBatch in the suite of synthetic datasets.

Ensemble − SeqB − NoiseY. In all the experiments we have conducted we kept the batch size $b$ and the number of batches $N$ for each dataset equal to the settings used in Section E.1. The neural network architectures are the same we have considered before, two layer networks with ReLU activations.

### E.2.1  OAE − DvD

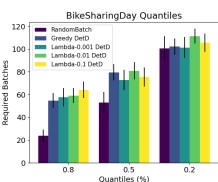

Figure 19: **NN 100-10**. $\tau$-quantile batch convergence performance of DetD vs RandomBatch in the BikeSharingDay dataset.

We implemented and tested the DetD algorithm described in Section C.0.1. In our experiments we set $\lambda_{\mathrm{div}} = 1$ and set $\widetilde{f}_t$ to be the result of solving the MeanOpt objective (see problem 3) for different values of $\lambda_{\mathrm{reg}}$. We set $\mathcal{K}$ to satisfy,

$$\mathbf{K}(a_1, a_2) = \exp\left(-\left\|\frac{a_1}{\|a_1\|} - \frac{a_2}{\|a_2\|}\right\|^2\right)$$

We see that across the suite of synthetic datasets and architectures (**NN 10-5** and **NN 100-10**) the performance of MeanOpt degrades when diversity is enforced (see figures 20 and 21 and compare with figures 8 and 9). A similar phenomenon is observed in the suite of UCI datasets (see figure 19 for DetD results on the BikeSharing dataset and figure 14 for comparison).

In contrast, we note that DetD beats the performance of RandomBatch in the A373, MCF7 and HA1E datasets and over the two neural architectures **NN 10-5** and **NN 100-10**. Nonetheless, the DetD is not able to beat RandomBatch in the VCAP dataset. These results indicate that a diversity objective that relies only on the geometry of the arm space and does not take into account the response values may be beneficial when $y_\star \notin \mathcal{F}$ but could lead to a deterioration in performance when $y_\star \in \mathcal{F}$. The algorithm designer should be careful when balancing diversity objectives and purely optimism driven exploration strategies, since the optimal combination depends on the nature of the dataset. It remains an interesting avenue for future research to design strategies that diagnose in advance the appropriate diversity/optimism balance for OAE − DvD.

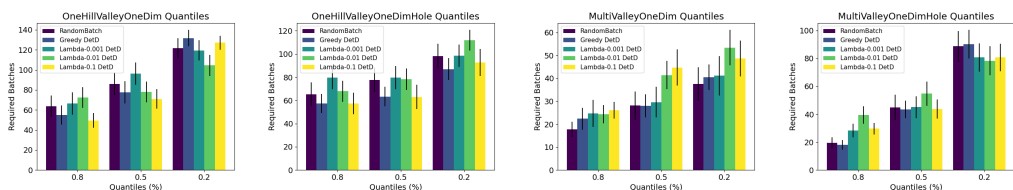

Figure 21: **NN 100-10**. $\tau$-quantile batch convergence performance of DetD vs RandomBatch in the suite of synthetic datasets.

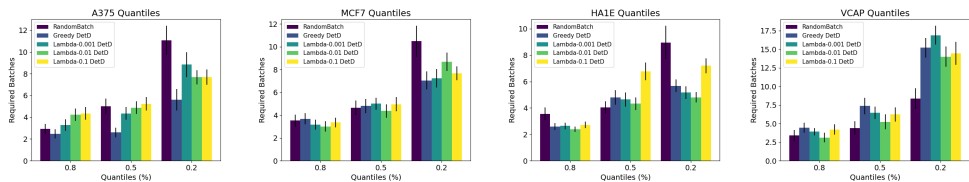

Figure 22: **NN 10-5**. $\tau$-quantile batch convergence performance of DetD vs RandomBatch in the suite of genetic perturbation datasets.

### E.2.2 OAE − Seq

In this section we present our experimental evaluation of the three tractable implementations of OAE − Seq we described in Section C.0.2. In our experiments we set the optimism-pessimism weighting parameter $\alpha = 1/2$ and the acquisition function to $\mathbb{A}_{\text{avg}}$. We are primarily concerned with answering whether 'augmenting' the MeanOpt, Ensemble and EnsembleNoiseY methods with a sequential in batch selection mechanism leads to an improved performance for OAE. We answer this question in the affirmative. In our experimental results we show that across datasets and neural network architectures adding in batch sequential optimism either improves or leads to no substantial degradation in the number of batches OAE requires to arrive at good arm.

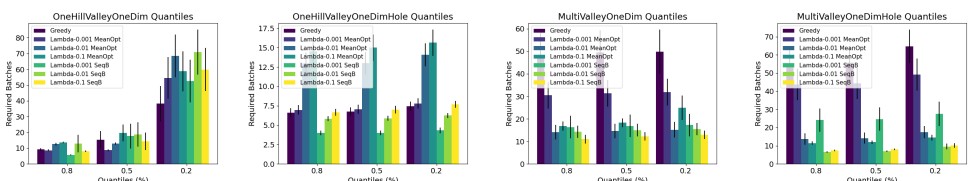

Figure 24: **NN 100-10**. $\tau$-quantile batch convergence performance of (**left**) MeanOpt vs SeqB, (**center**) Ensemble vs Ensemble − SeqB and (**right**) Ensemble − NoiseY vs Ensemble − SeqB − NoiseY on the suite of synthetic datasets.

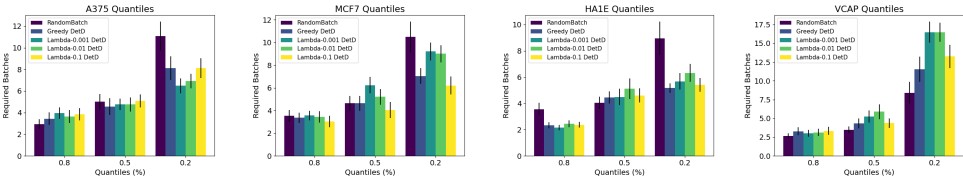

Figure 23: **NN 100-10**. $\tau$-quantile batch convergence performance of DetD vs RandomBatch in the suite of genetic perturbation datasets.

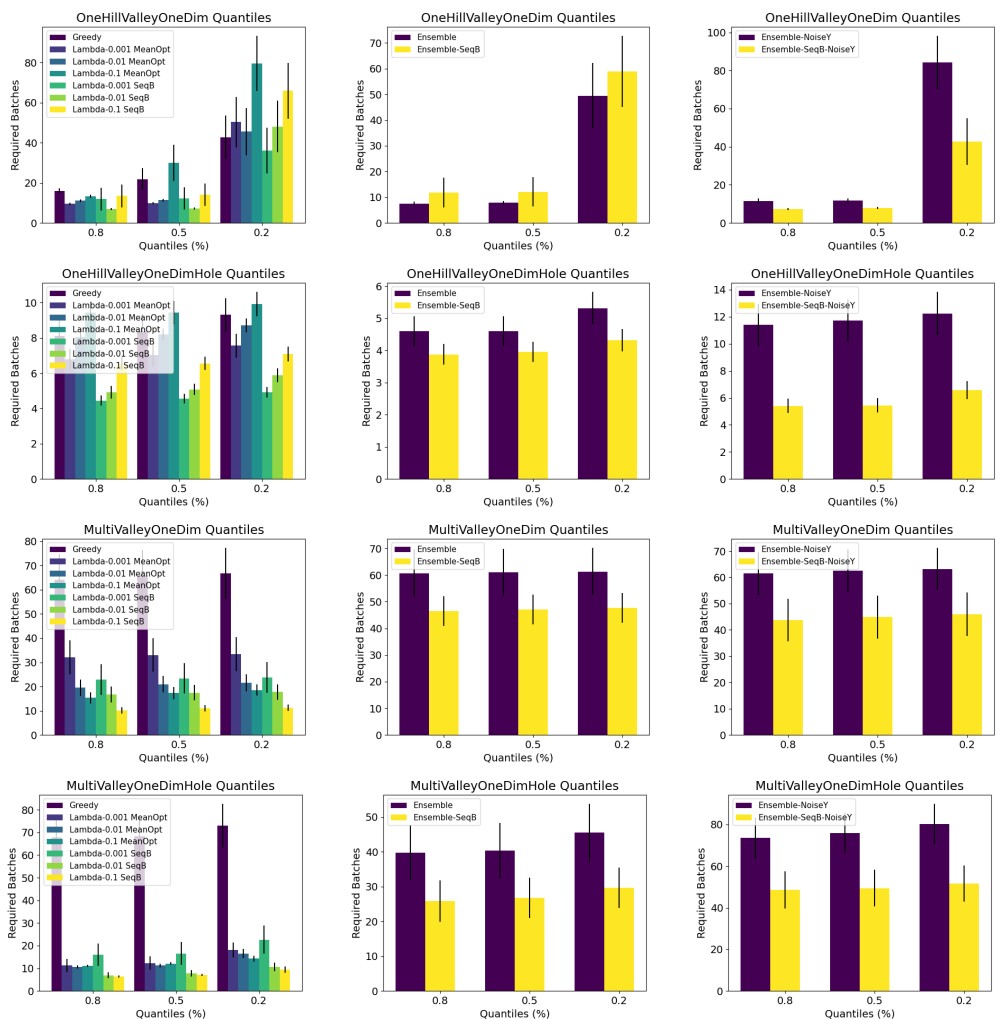

Figure 26: **NN 10-5**. $\tau$-quantile batch convergence performance of (**left**) MeanOpt vs SeqB, (**center**) Ensemble vs Ensemble − SeqB and (**right**) Ensemble − NoiseY vs Ensemble − SeqB − NoiseY on the suite of synthetic datasets.

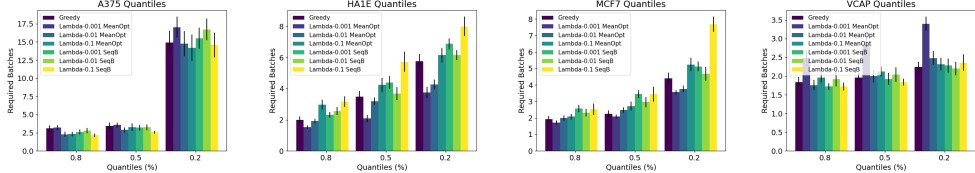

Figure 25: **NN 100-10**. $\tau$-quantile batch convergence performance of MeanOpt vs SeqB on the gene perturbation datasets.

More precisely in figures 26 and 24 we show that in the set of synthetic datasets adding a sequential in batch selection rule improves the performance of OAE across the board for (almost) all datasets and all methods MeanOpt, Ensemble and EnsembleNoiseY and two neural architectures **NN 10-5** and **NN 100-10**. Similar gains are observed when incorporating a sequential batch selection rule to OAE in the BikeSharingDay and BikeSharingHour datasets when F corresponds to the class **NN 100-10** (see figure 27). Finally, we observe the performance of OAE − Seq either did not degrade or slightly improved that of MeanOpt, Ensemble and EnsembleNoiseY in the BlogFeedback and the genetic perturbation datasets (see figures 34, 25 and 38). We conclude that incorporating a sequential batch decision rule, although it may be computationally expensive, is a desirable strategy to adopt. It

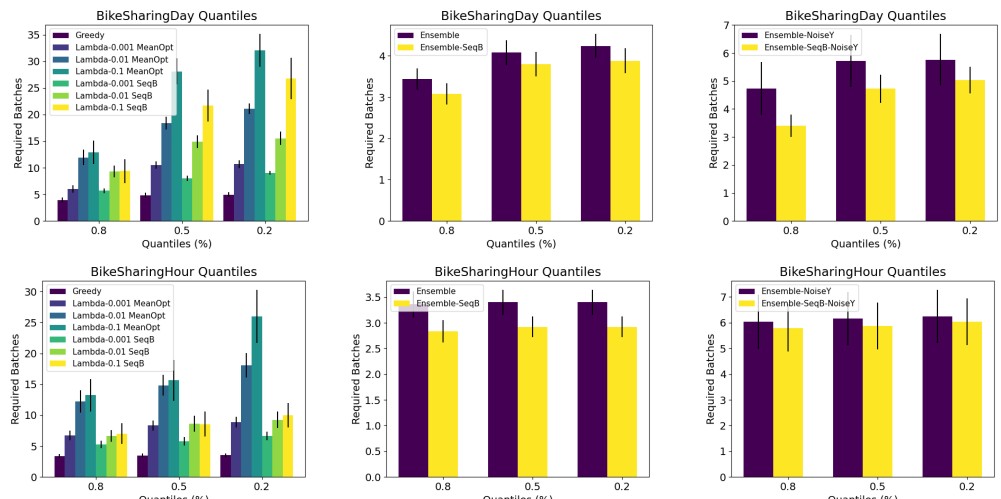

Figure 27: **NN 100-10**. $\tau$-quantile batch convergence performance of (**left**) MeanOpt vs SeqB, (**center**) Ensemble vs Ensemble − SeqB and (**right**) Ensemble − NoiseY vs Ensemble − SeqB − NoiseY on the BikeSharingDay and BikeSharingHour datasets.

is interesting to note that in contrast with DetD the diversity induced by SeqB did not alleviate the subpar performance of MeanOpt in the set of genetic perturbation datasets. This can be explained by SeqB induces query diversity by fitting fake responses and it therefore may be limited by the expressiveness of $\mathcal{F}$. The DetD instead injects diversity by using the geometry of the arm space and may bypass the limitations of exploration strategies induced by F. Unfortunately DetD may result in suboptimal performance when $y_\star \in \mathcal{F}$.

## F  CELLULAR PROLIFERATION PHENOTYPE

Let $\mathcal{G}$ be the list of genes present in CMAP also associated with proliferation phenotype according to the Seurat cell cycle signature, and let $x_{i,g}$ represent the level 5 gene expression of perturbation $a_i$ for gene $g \in \mathcal{G}$. We define the proliferation reward for perturbation $a_i$ as the average expression of the genes in $\mathcal{G}$,

$$f_*^{\text{prolif}}(a_i) = \frac{1}{|\mathcal{G}|} \sum_{g \in \mathcal{G}} x_{i,g}$$

For convenience, $\mathcal{G} = \{$AURKB, BIRC5, CCNB2, CCNE2, CDC20, CDC45, CDK1, CENPE, GMNN, KIF2C, LBR, NCAPD2, NUSAP1, PCNA, PSRC1, RFC2, RPA2, SMC4, STMN1, TOP2A, UBE2C, UBR7, USP1$\}$.

## G    FURTHER EXPERIMENTS

### G.1    SYNTHETIC DATASETS

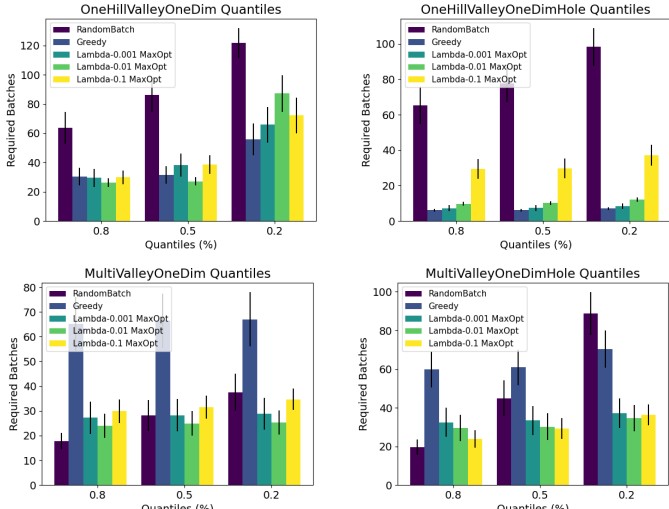

Figure 28:   **NN 10-5**. MaxOptimism comparison vs RandomBatch over the suite of synthetic datasets.

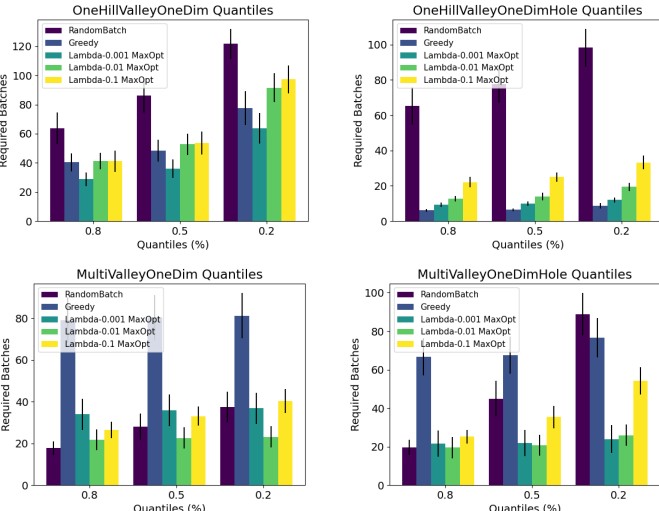

Figure 29: **NN 100-10**. MaxOptimism comparison vs RandomBatch over the suite of synthetic datasets.

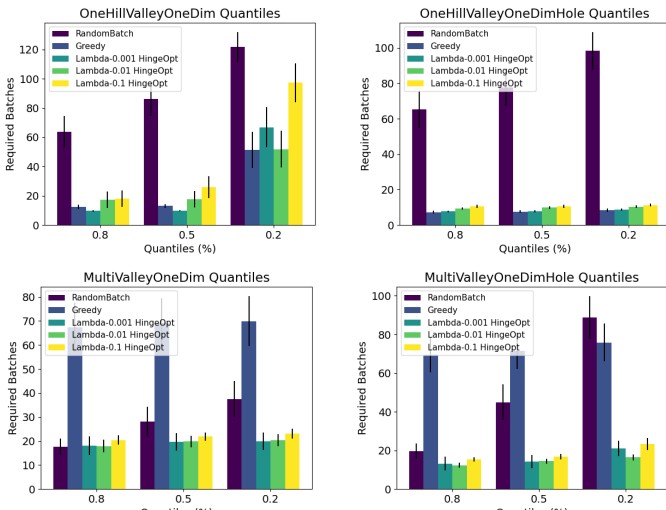

Figure 30: **NN 10-5**. HingePNorm comparison vs RandomBatch over the suite of synthetic datasets.

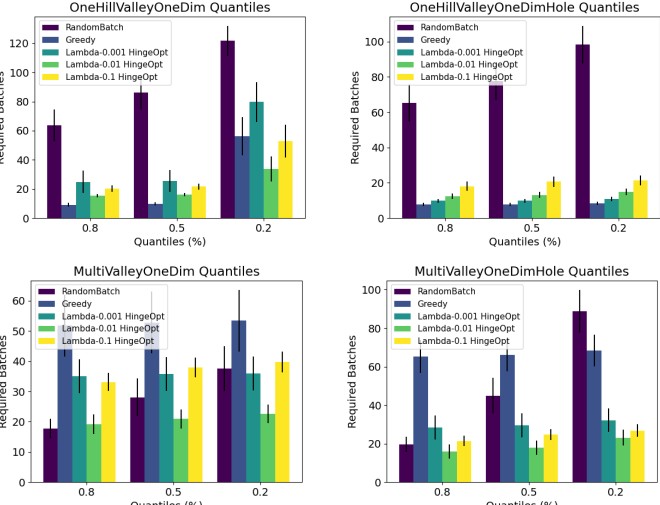

Figure 31: **NN 100-10**. HingePNorm comparison vs RandomBatch over the suite of synthetic datasets.

## G.2 REGRESSION FITTED DATASETS

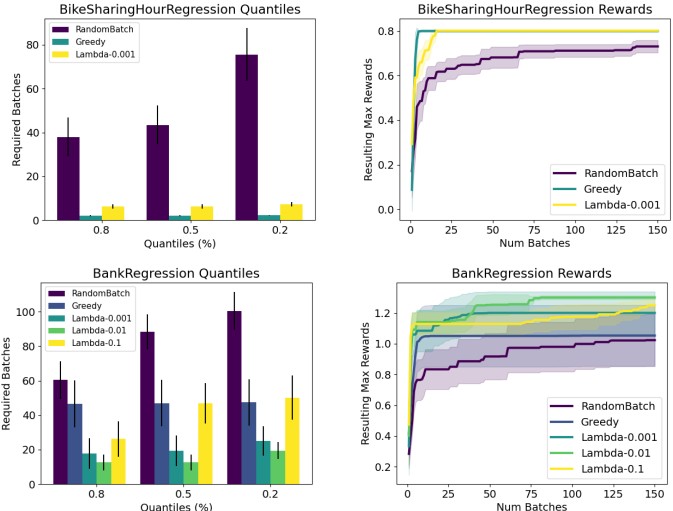

Figure 32: **NN 100-10**. BikeSharingHour and Bank Regression Fitted Datasets. Our algorithms perform substantially better than RandomBatch. Optimistic approaches are better than Greedy in the Bank dataset.

## G.3 UCI PUBLIC DATASETS

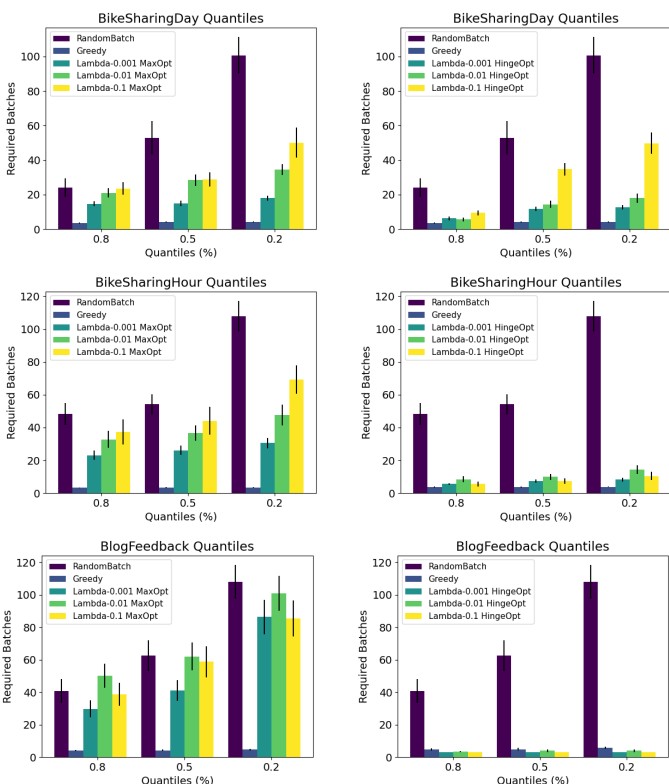

Figure 33: **NN 100-10**. MaxOptimism and HingePNormOptimism comparison vs RandomBatch in the BikeSharingDay, BikeSharingHour and BlogFeedback datasets.

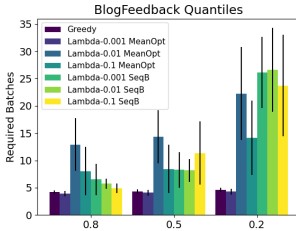 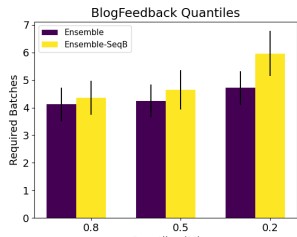 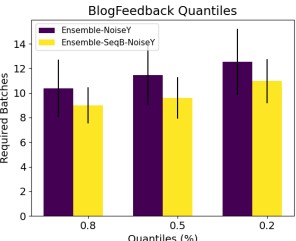

Figure 34:    **NN 100-10**.    $\tau$-quantile batch convergence performance of (**left**) MeanOpt vs SeqB, (**center**) Ensemble vs Ensemble − SeqB and (**right**) Ensemble − NoiseY vs Ensemble − SeqB − NoiseY on the BlogFeedback dataset.

### G.4 Transfer Learning Biological Datasets

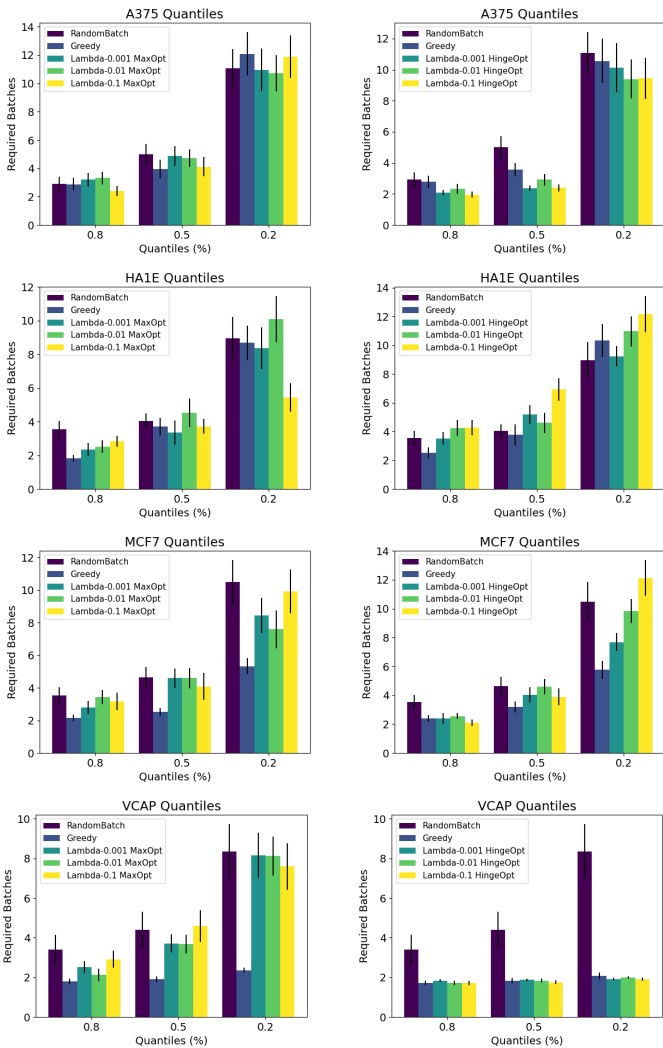

Figure 35: **NN 10-5**. MaxOptimism and HingePNormOptimism comparison vs RandomBatch.

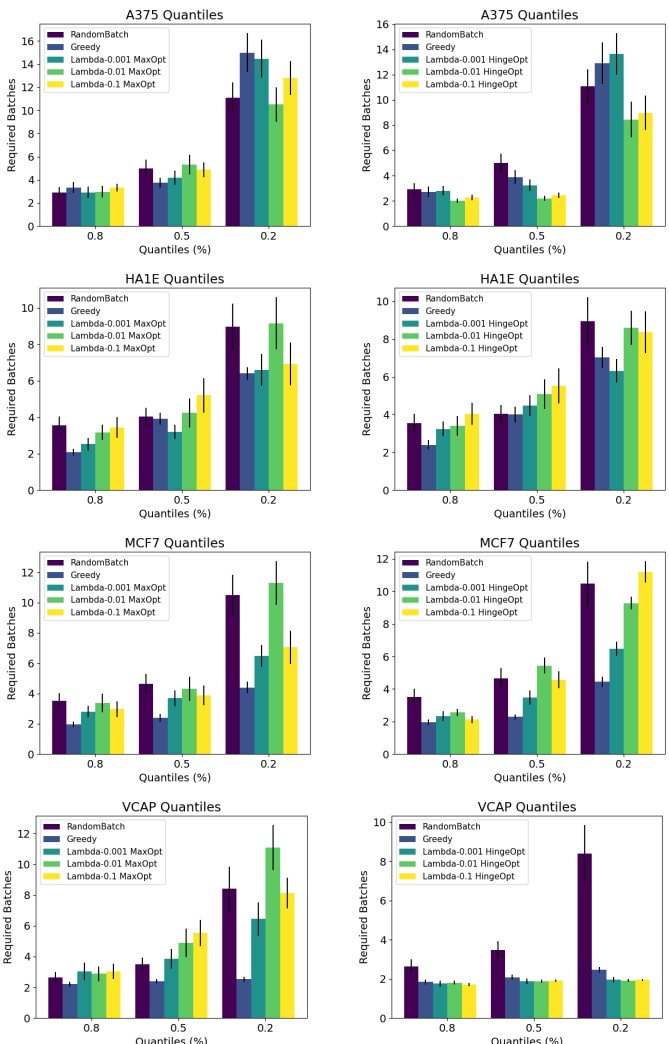

Figure 36: **NN 100-10**. MaxOptimism and HingePNormOptimism comparison vs RandomBatch.

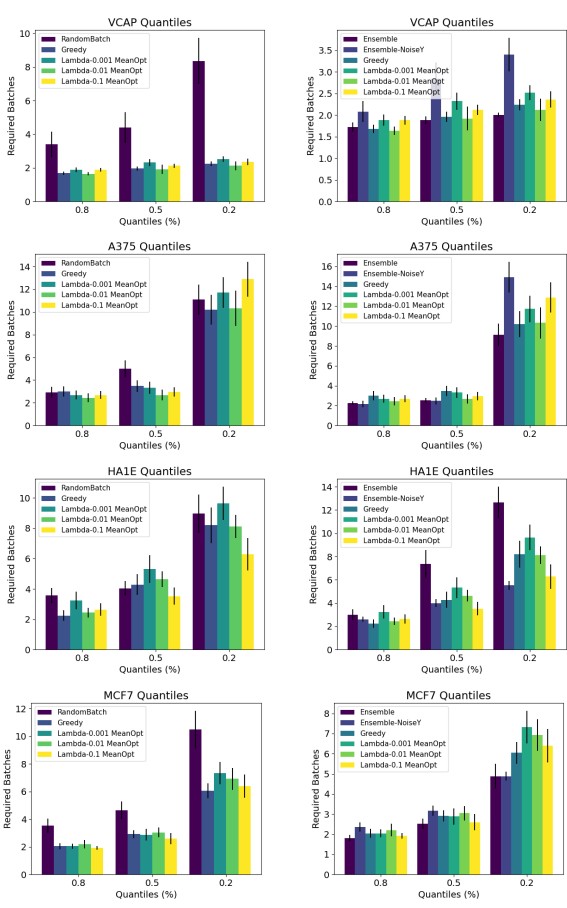

Figure 37: **NN 10-5**. $\tau$-quantile batch convergence of MeanOpt (**left**), Ensemble and EnsembleNoiseY (**right**) on genetic perturbation datasets.

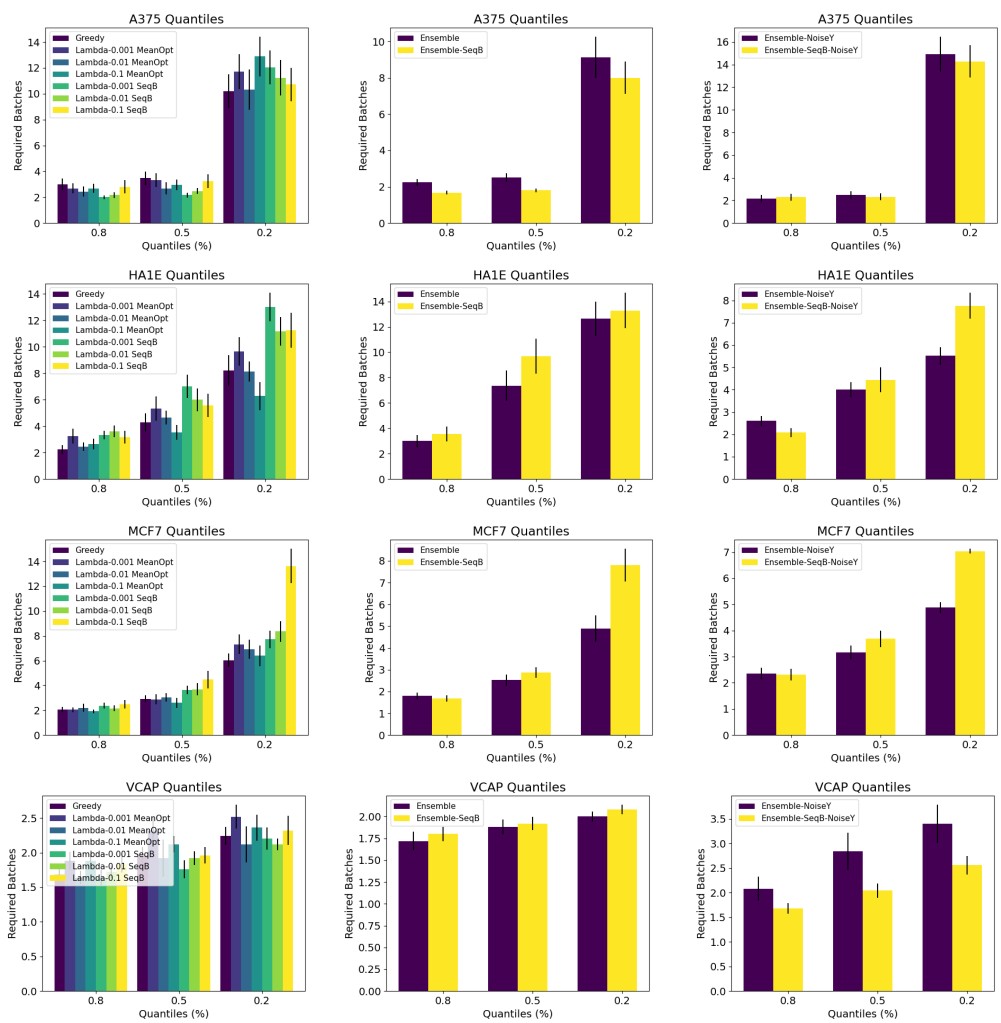

Figure 38: **NN 10-5**. $\tau$-quantile batch convergence performance of (**left**) MeanOpt vs SeqB, (**center**) Ensemble vs Ensemble $-$ SeqB and (**right**) Ensemble $-$ NoiseY vs Ensemble $-$ SeqB $-$ NoiseY on the genetic perturbation datasets.

