# OpenReview forum: "Neural Design for Genetic Perturbation Experiments"
_ICLR.cc/2023/Conference — ICLR 2023 notable top 25%_

### Official Review · Reviewer_bZbX · 2022-10-22

**Confidence:** 3
**Correctness:** 3
**Technical Novelty And Significance:** 2
**Empirical Novelty And Significance:** 2
**Recommendation:** 6

**Clarity, Quality, Novelty And Reproducibility:**

First sentence of the abstract is not clear to the average ML reader. What is CAR-T, CAR-NK, CAR-NKT? As a trained ML bioinformaticist, I don’t even know the difference between those three.

Again for the average ML reader, what is shRNA or CRISPR? What is a genetic knockout? There are no references in the first paragraph of the introduction for the reader to continue reading from.

Section 1 - “​​Our experimental evaluation covers both neurally realizable and not neurally realizable function landscapes.” What does this mean?

Section 2 - “We provide guarantees for the no noise setting we study based on the Eluder dimension…” What does this mean?

Section 5 - Can we have some reference for the cell lines described VCAP, HA1E, MCF7, A375?

In Figure 2 - Where are the exact experimental details of the figure legends for MeanOpt, Greedy, and Lambda0.XX- ? It is unclear how these abbreviations tie into the methods described, and the reader should not have to hunt through the supplement to look this up.

In Figure 5, what is the variability around those estimates?

It is unclear how the conclusions described at the bottom of page 8 relates to Figure 4. For example, what does “thus indicating this function class is too far from the true responses values for A375” mean? What function class are you referring to? Though there are scores of figures and descriptions in the appendix, the important points to convince me, the reader, should be able to be summarized in the main work.

The metric you are optimizing differs for the two datasets, which makes comparison difficult.


**Strength And Weaknesses:**

Very interesting problem, and not a lot of others in this space–great to define this problem concretely!

A wide array of related work is outlined and organized, not only with section headings, but also citations.

My difficulty with theory-heavy biology paper lies in statements like this: “the learner is required to query a batch of ..arms…and observe noiseless responses…In the setting of neural perturbation experiments the responses are the average of many expression values across a population of cells, and thus it is safe to assume the observed response is almost noiseless.” As someone who has analyzed dozens of these datasets, this is most certainly not true. In real world “shRNA and CRISPR” experiments, there is always variability, minimally in knock-down or edit level, phenotypic response, experimental design, etc. Though a population of cells is sampled, there will always be variability. Since you have datasets, can you characterize the experimental noise of a given knockout? If you don’t have this data, you cannot make this assumption.

To what extent can the uncertainty estimates for your model be used? How were they calibrated, and how much poor calibration can you tolerate? For proper uncertainty estimates, why not Gaussian Processes as a baseline, or maybe even Bayesian Neural Networks? Why are only Ensemble methods described in Section 4.1.2 as the only method to create uncertainty, but then Bayesian Neural Networks are used in Section 5.0.1? Why were those not described in Section 4?

In Figure 1, and that associated analysis, to what extent do we care about cross-cell-line variability? I think showing this generalization is useful. However, how much do we generalize across perturbations?

The use of a VAE to represent sequences is concerning to me. Why don’t just input raw data into a neural network? By using the VAE representations, you are implicity assuming you can reasonably represent all variability in expression. If a series of genes, or gene networks, were removed during training of the VAE, and projected with a learned model, how are they represented? If these KOs are incorporated into both the VAE and neural net training, can you still generalize to those predictions?


**Summary Of The Paper:**

The authors define new methods to select batches of perturbations of cells that are most likely to inform and optimize on a phenotypic metric of interest.

**Summary Of The Review:**

I think that the problem the authors are trying to solve is very interesting, and I applaud them on their work. However, I found that this paper lacked focus and refinement–important points should fit into the main text. Moreover, I found there to be a number of arbitrary design decisions with VAEs and uncertainty estimate models that detracted from the core contributions stated by the author.

---

> ### Author Response · Authors · 2022-11-11
> **Response to reviewer bZbX**
>
> We want to thank the reviewer for their kind and detailed comments and would really like to encourage the reviewer to revise their score and engage in a conversation with us the authors during the rebuttal period. We hope the reviewer read the ‘response to all reviewers’ we posted at the top of the page and would love to address any further questions they may have about our work. We are very happy to have a back and forth and will be fast at replying to the reviewers questions if any.
>
> Thank you so much for your insightful comments. As the reviewer has noted we take the view that for this specific problem domain the zero noise is an appropriate modeling assumption. We very much recognize the reviewer’s concern here and would like to address it specifically. Our algorithms have simple noise robust versions (simply run each experiment some number of times roughly equal to $1/\epsilon^2$ to learn the experiment's mean value up to $\epsilon$ accuracy, or at least learn the noise’s variance). Although this will not be a theoretically optimal strategy to deal with noisy responses, it should work well in practice. We actually believe the zero noise setting is interesting in itself as a research direction. Optimizing an unknown function over a set of vectors is already a challenging problem. Characterizing the statistical complexity of optimization in this setting (with exact function evaluation) is a surprisingly neglected but fundamental problem. We believe our theoretical results make interesting inroads towards this. The design of algorithms for the noisy query model has to deal with two issues, the optimization problem over the function class and action set, and the complexities of noisy evaluation. We think that making inroads towards the first problem in isolation is a good starting point to understand these problems in a more systematic way.
>
> Other baselines. The main issue we found here was compute. These methods require significantly more compute than the ones we have proposed in this paper, and unfortunately we have a limited amount of compute. We really hope our methods can be adopted by the community and further benchmarked against these and other alternatives.
>
> Figure 1 and VAEs. This is what figure 1 is trying to show. figure 1 shows the evolution of the regression loss of fitting all the responses for the different datasets when the gene-features are built via an auto-encoder (on the 978 dimensional gene expression space) using the VCAP cell line (recall this work is on optimizing single-gene knockout perturbations). The architecture used is exactly the same one as used in the decoder. As we explain in the paper, this shows that using features learned this way and transferring them to a new cell line successfully captures the responses of some of the cell lines (VCAP, HA1E and MCF7) but not all (A375). This means the features for A375 and therefore the NN function classes we use to fit it are most likely not able to capture exactly the true responses for this cell line (i.e. they are misspecified). This is a very natural phenomenon that may occur in practice. We show in our experiments that despite this that for example OAE-DvD (DetD) is able to perform well in this misspecified dataset. In section E.2.1 we provide a discussion of this phenomenon.

---

> > ### Author Response · Authors · 2022-11-11
> > **continuation**
> >
> > Clarity. Thanks so much for these comments. They are very useful. We will add references so the reader can investigate the meaning of all the bio terms. We really appreciate this comment.  “​​Our experimental evaluation covers both neurally realizable and not neurally realizable function landscapes.” - this means landscapes where the function class ($\mathcal{F}$) truly captures the response and landscapes where this function class cannot fit the true response. “We provide guarantees for the no noise setting we study based on the Eluder dimension…”  The eluder dimension is a statistical dimension used to characterize the complexity of some online learning problems. Our theoretical guarantees are in terms of this fundamental quantity. Figure 2. We respectfully disagree with the reviewer here. The description of Greedy, MeanOpt is in section 4.1.1. In section 4.1 all the tractable implementations of our algorithmic principles are described. We will add a short description specifying how the Lambda xx MeanOpt labels in the figures means we are running the MeanOpt algorithm introduced in section 4.1.1 with a value of lambda equals to xx.  Figure 5. Thanks so much for this comment. We will add the standard deviation values to the revision. The reviewer indicates this phrase is unclear “thus indicating this function class is too far from the true responses values for A375”. This means the NN or model we are using cannot fit the true response for A375. This is saying that if you look at figure 4, the losses for A375 are substantially higher than those for any other cell line. This means the fit is imperfect for all the function classes (NN architectures) for this cell line. This, as we have explained previously in the paper, is because the features we have used (derived from the VAE in VCAP) are ‘misspecified’ for cell line A375. We think these are very interesting observations because they match what we would expect to see in practice. That is, if we use a feature set derived from a related task, it is not clear if this feature map will work well. Since this is a common occurrence in practical domains, it is important to test our algorithms under those conditions. This is the purpose of this discussion / plots.

---

> > > ### Author Response · Authors · 2022-11-14
> > > **Follow up**
> > >
> > > Hello,
> > >
> > > We would like to ping the reviewer to check if they have read our rebuttal and if they have any more questions. We would be happy to reply.
> > >
> > > The authors

---

> > > > ### Comment · Reviewer_bZbX · 2022-11-17
> > > > **Follow up**
> > > >
> > > > Thank you for your follow up and revisions to the work. I think the revisions make the paper more clear to the reader.
> > > >
> > > > However, many of the fundamental assumptions and experiments of the paper remain the same, and I feel my score still accurately reflects this work.

---

> > > > > ### Author Response · Authors · 2022-11-17
> > > > > **Thanks and a bit more information**
> > > > >
> > > > > We want to thank the reviewer. Their suggestions have been extremely helpful. We would nonetheless like to make another attempt at explaining our zero noise assumption in case this is part of the "fundamental assumptions" mentioned in this last response that makes the reviewer reticent to increase their score.
> > > > >
> > > > > We want to thank the reviewer for their thoughtful feedback. We want to emphasize that just as our title implies the main application we are trying to target in this work is that of genetic perturbation experiments. We believe this is a valid model of this problem setting. Our zero noise assumption is motivated by real-world applications of active learning methods like ours in which selecting the same arm multiple times is almost never done because the noise of that arm is very low relative to the cost of selecting it. In settings like these, arms are almost never selected more than once. In the genetic perturbation context, our reward function is already the average response across technical replicates, each of which is itself an average response across the technical batch of cells (in a single well of a plate) receiving the same genetic perturbation. This average of averages motivates our zero-noise assumption.
> > > > >
> > > > > We agree with the reviewer that a further direction outside the scope of this work would be to explore the noise introduced by biological replicates, for example performing a perturbation from additional human primary cell donors. Neither the CMAP dataset nor the GeneDisco datasets offer the option of observing the same genetic perturbation on a new donor, but as richer genetic perturbation datasets become publicly available across many (10+) donors, we will be excited to extend our approach to accommodate this additional inter-donor source of variability.
> > > > >
> > > > > In any case we are really pleased the reviewer agrees about how interesting is the problem we are studying in this work! Moreover, we really would like to point out that despite our work being focused on the biological aspect of this problem, our theoretical results are highly non-trivial and could have stood as the basis of a single theory paper on their own. We explain why below:
> > > > >
> > > > >  First, recall that in the stated theorem (Theorem 4.1) the regret upper bound scales as $\mathcal{O}\left( d b + \omega \sqrt{d}
> > > > > T b \right)$. We want to draw attention to the square root dependence on $d$  in the linear misspecification component of the regret. This is very significant because it generalizes the results of [1] to the setting of function approximation (beyond linear) and when the number of queries per round equals $b$, possibly more than $1$. The main ingredient in our proof (Proposition D.4) is a completely novel result. It is \emph{not} a simple rehashing of Proposition 4 in [2] but a unique result on its own. We encourage the reviewer to compare it with the results in [2] to see how the proof results and techniques are different. In particular, the proof of Proposition D.4 involves a careful and novel use of Hall’s marriage theorem in this setting. Moreover, Proposition D.4 can be used to show the regret in noisy settings and for example when $\omega = 0$, scales as $\mathcal{O}\left( d \sqrt{Tb} + bd \right)$ this again is significant because a naive analysis of a parallel non linear bandit procedure would scale as $\mathcal{O}\left( d b \sqrt{T}\right)$ (linearly with $b$). Instead, with Proposition D.4 we can show the parallel regret scales as the sequential regret of running Tb sequential queries (i.e. the b dependence is of order $\sqrt{b}$) up to a burn in factor of order db. This can be done by simply substituting our novel result (Proposition D.4) into the proof of Lemma 2 in [2]. In other words, the cost of parallelism only emerges at the beginning. This generalizes for example all the results of [3] since it would yield results for parallel exploration (b  > 1) with nonlinear functions.  \textit{Parameter free} This is indeed true, and it is a major issue in \emph{many} bandit algorithms. In this case this can be mitigated by simply using an online model selection wrapper around our algorithm, which for selecting the misspecification parameter will achieve near optimal oracle rates.
> > > > >
> > > > > [1] Learning with Good Feature Representations in Bandits and in RL with a Generative Model.
> > > > >
> > > > > [2] Eluder Dimension and the Sample Complexity of Optimistic Exploration.
> > > > >
> > > > > [3] Parallelizing Contexual Linear Bandits.

---

### Official Review · Reviewer_s1ZP · 2022-10-23

**Confidence:** 4
**Correctness:** 4
**Technical Novelty And Significance:** 4
**Empirical Novelty And Significance:** 4
**Recommendation:** 8

**Clarity, Quality, Novelty And Reproducibility:**

High marks on clarity, quality and novelty (see above). I am not sure if the code is or will be shared. Please clarify.


**Strength And Weaknesses:**

The paper is generally well-written (a few minor English errors should be easy to fix) and clear. The proposed method appears novel (although inspired by much previous work, and an extensive set of relevant papers are cited) and the results appear promising. The paper also includes a bandit theoretical results bounding the regret.

It is clear that the authors have thought carefully about many issues arising in the context of their targeted application (such as diversity within each batch, how to evaluate performance).

The main weakness is that there should have been more comparisons against existing methods rather than against a single strawman (the GeneDisco public implementation).

The paper focuses on the single-intervention case and the algorithms would not scale to compositions of interventions (e.g. multi-gene interventions) because the different actions must be enumerated and scored separately. See the recent work on GFlowNets (NeurIPS 2021, ICML 2022) as a possible way to train a policy to generate actions in a way that favors a diversity of candidates in each batch, as an alternative to explicit screening of all the possible actions.

I am concerned with the use of the inf-norm in 4.3 because a hugely diverse set of \tilde{y}* could minimize it.

I would like to see something added to explain how to select lambda_reg (which controls the trade-off between fitting the data and maximizing predictions, i.e., the weight uncertainty in the acquisition function) and how to do it without cheating (e.g. to get the results in Fig. 5), i.e., without using the downstream results of the active learning iterations.

More references should be added regarding the related work in uncertainty estimation without an explicit Gaussian posterior (such as available with GPs), i.e., with neural networks. See e.g. DEUP (Lahlou et al 2021).

**Summary Of The Paper:**

This paper introduces a novel bandit / active learning method tailored to the problem of single-gene interventions in cell biology experiments, where we can have batches of parallel interventions and only a few rounds of active learning are feasible. The new bandit method (Optimistic Arm Elimination or OAE) is based on either an existing uncertainty estimator or a novel optimization objective for the reward predictor that trades-off between fitting the data and maximizing the predictions. A diversity regularizer can be added to regularize each batch to be somewhat diverse. A large set of experiments to compare the different variants under study is performed, along with one comparative experiment (on 4 datasets) on the GeneDisco benchmark (using the public implementation of GeneDisco).


**Summary Of The Review:**

This paper is clear and describes a strong, novel and well-thought out work on bandit methods in cases where very few rounds of batched interventions are possible. It could be stronger with more points of comparison, i.e., against more of the existing state-of-the-art methods.

---

> ### Author Response · Authors · 2022-11-11
> **Response to reviewer s1ZP**
>
> We want to thank the reviewer for their kind and detailed comments and would really like to encourage the reviewer to engage in a conversation with us the authors during the rebuttal period. We hope the reviewer read the ‘response to all reviewers’ we posted at the top of the page and would love to address any further questions they may have about our work. We are very happy to have a back and forth and will be fast at replying to the reviewers questions if any.
>
>
> We are really excited the reviewer had a great time reading our work. We thought very hard about how to evaluate performance and are very proud of the different diversity inducing methods we devised. We are happy the reviewer is too.
>
> References and Comparison. Thanks so much for the references on GFlowNets, DEUP and others. We will certainly expand our related work section. We will add these to our discussion. As for the GenDisco comparison, we thought the best way to  compare against a publicly available, accepted implementation. We thought this was better than any ad-hoc comparison with other methods. This is also why in our other experiments we compare our methods against simple uncontroversial algorithms like RandomBatch, with the objective to show that our methods indeed produce gains. We hope there is a push in the community to have better, larger and easier to expand public methods libraries (such as GeneDisco). We think it would be very exciting to contribute towards that.
>
> Inf-norm. We are slightly confused about the inf-norm comment. We define the distance between $\mathcal{F}$ and the $y_\star$ responses as $\min_{f \in \mathcal{F}} \max_a | f(a) - y_\star(a)|$ so that this distance can only be small if there is an $f$ that is uniformly (for all $a$) close to $y_\star$.
>
> Lambda-reg. The easiest way to do this is via an online model selection algorithm. In this setting one would initialize multiple algorithms with different values of lambda, and have a special bandit algorithm select among the different models (algorithms with different values of lambda) in an online way. Doing this should ensure the performance of the resulting lambda selection rule is competitive vs the best lambda that could have been selected in hindsight.

---

> > ### Comment · Reviewer_s1ZP · 2022-11-11
> > **Response to rebuttal; lambda-reg and comparisons**
> >
> > Thank you for the rebuttal, but it did not completely satisfy me.
> >
> > Comparisons: aren't there other published methods that could be applied there and that you could have compared with?
> >
> > Lambda-reg: your answer is about what *could* be done in obtain a proper selection of lambda-reg. But what did you *actually* do for your experiments?

---

> > > ### Author Response · Authors · 2022-11-11
> > > **Follow up**
> > >
> > > In our experiments we ran multiple values of lambdas (we do not have a specific lambda selection algorithm). We want to reiterate this is an algorithm hyper-parameter. This is the exact same case as in many other algorithms that have an exploration controlled hyper-parameter (think of a scaling, or kernel choice in GPs, or the width of confidence radii in bandit algorithms) and thus typically not a cause of major concern as long as multiple hyperparameter choices are robust. And as you can see from our plots our algorithms (in particular MeanOpt) are fairly robust to what lambda value we choose. We agree with the reviewer that it is an exciting question to figure ways of adapting to this value online. In our GeneDisco OAE-Bayesian implementation we set the effective lambda value to $1$. We did not make an effort here to pick a 'good' parameter value and we still obtained good results. Other baselines we could have compared are methods that use Gaussian Processes such as GP − BUCB. Unfortunately we do not have access to a public implementation of this algorithm and its computational requirements (GPs are notoriously computationally intensive) would have made it impossible for us, having limited computational resources, to run. We really made an effort (both in engineering and cost) to have as thorough of an experiment suite (see our 38 experimental results figures with many many different methods ranging from MeanOpt, MaxOpt, HingeOpt, OAE-DvD, OAE-Seq, etc ...) as possible. We really really hope this effort is evident in our manuscript.

---

### Official Review · Reviewer_UfWC · 2022-10-24

**Confidence:** 3
**Correctness:** 3
**Technical Novelty And Significance:** 3
**Empirical Novelty And Significance:** 2
**Recommendation:** 6

**Clarity, Quality, Novelty And Reproducibility:**

The introduction is clear but I found that the method section was too long and the results too short for my tastes. This is subjective, of course. Also the figures were too small and difficult to interpret, which was a bigger issue.

The method seems novel although i am not an expert in the area. The results appear to be reproducible.

**Strength And Weaknesses:**



Summary
The authors investigate the problem of predicting how to perturb cells via genetic manipulations in order to shape their phenotype in a desired way. Their approach to this problem is to rely on a novel form of batch query bandit optimization, their Optimistic Arm Elimination (OAE) principle. They use their OAE to show that they can often plan effective experiments in fewer tries than baseline methods.


Strengths
- The problem setting is extremely interesting, and I believe holds huge potential for drug discovery. Also, this is a problem that I believe has been under studied because it is so difficult to model.
- The authors test their method on a variety of datasets.

Weaknesses
- I'm not sure what the purpose of Figure 1 is. It's showing regression loss of different models trained on a few cell lines. The model hasn't converged. There's no test performance. What should we take from this.
- What motivated the model designs? The authors switch between NN 1500-300 to NN100-10 to NN10-5 without much explanation. What's going on here?
- Figure 3 is above figure 2 — confusing.
- Results are extremely sparse and confusing. MeanOpt sometimes requires fewer batches than random or greedy and sometimes doesn't. On the GeneDisco experiments it looks like the author's models work the best on 3/4 conditions but I have no idea of effect size.
- The authors acknowledge that they assume noiseless responses and that they will focus on noisy responses in the future. For biological applications -- and especially for any kind of sequencing -- I'm worried that the noiseless assumption is overly optimistic even when operating on populations. I'd like to hear more from the authors on why an assumption of no noise is OK here.

Minor

- The citations in text do not always appear correctly (I think you should be using \citep sometimes when you are not).
- The model names are very confusing. A figure might help? Also easier to grok names? They're all the same type of model with different widths (right?), so something like narrow, baseline, wide might help.

**Summary Of The Paper:**

The authors investigate the problem of predicting how to perturb cells via genetic manipulations in order to shape their phenotype in a desired way. Their approach to this problem is to rely on a novel form of batch query bandit optimization, their Optimistic Arm Elimination (OAE) principle. They use their OAE to show that they can often plan effective experiments in fewer tries than baseline methods.


**Summary Of The Review:**

The problem setting is incredibly interesting and exciting but I thought the paper was difficult to read and the results not all that compelling. I am also worried that the key assumption of no-noise is fundamentally flawed for biological data, so I'm curious what the authors and other reviewers think about that. I think there's great potential here, however, as it stands, I am leaning towards reject but still open to changing my review during the discussion period.

---

> ### Author Response · Authors · 2022-11-11
> **Response to reviewer UfWC**
>
> We want to thank the reviewer for their kind and detailed comments and would really like to encourage the reviewer to revise their score and engage in a conversation with us the authors during the rebuttal period. We hope the reviewer read the ‘response to all reviewers’ we posted at the top of the page and would love to address any further questions they may have about our work. We are very happy to have a back and forth and will be fast at replying to the reviewers questions if any.
>
> We will address the reviewers perceived weaknesses one by one:
>
> 1) As explained on page 8, figure 1 shows the evolution of the regression loss of fitting all the responses for the different datasets when the features are built via an auto-encoder using the VCAP cell line. The architecture used is exactly the same one as used in the decoder. As we explain in the paper, this shows that using features learned this way and transferring them to a new cell line successfully captures the responses of some of the cell lines (VCAP, HA1E and MCF7) but not all (A375). This means the features for A375 and therefore the NN function classes we use to fit it are most likely not able to capture exactly the true responses for this cell line (i.e. they are misspecified). This is a very natural phenomenon that may occur in practice. We show in our experiments that despite this that for example OAE-DvD (DetD) is able to perform well in this misspecified dataset. In section E.2.1 we provide a discussion of this phenomenon.
>
> 2) The NN architecture 1500-300 for the autoencoder was chosen via trial and error to ensure the regression loss on VCAP (where we learn the features) is low and the features were informative. The network architectures we have used in our experiments are meant to show the performance of our algorithms as the architecture is varied from linear to small (NN10-5) to medium (NN100-10) to large (NN300-100). Our computational resources are limited so we could not possibly run a larger sweep over more architectures.
>
> 3) Figure labeling we will fix this.
>
>
> 4) We are not claiming a single method will work over-the-board for all datasets. What we did try to do is to provide intuition for what methods may work for a given dataset, and why. We have explained in detail why we think a certain algorithm may perform better than others depending on the nature of the dataset, for example, the shape of the responses in the one dimensional datasets, or the quality of the features in our biological experiments. We in fact think of this as a contribution of our work, covering several situations a practitioner may encounter and providing intuition of what methods are best for any given situation. We also respectfully disagree with the term sparse used here. We have 38 figures showing experimental results. These cover all the methods we have introduced, on all the datasets we are testing on and with different neural networks (and even linear models). Figure 5. Thanks so much for this comment. We will add the standard deviation values to the revision.
>
> 5) No noise.  We would like to respectfully push back on the reviewers’ language here. Studying the zero noise setting was an active decision as opposed to a ‘fallback’ option as the reviewer seems to imply. This is because in the setting of genetic perturbation experiments the experiment’s feedback is an average over a population of cells. We take the view that for this specific problem domain the zero noise is an appropriate modeling assumption. On this note, our algorithms have simple noise robust versions (simply run each experiment some number of times equal to $1/\epsilon^2$ to learn the experiments mean value up to $\epsilon$ accuracy). We leave more refined strategies for the noisy setting as future work. Moreover, we actually believe the zero noise setting is interesting in itself as a research direction. Optimizing an unknown function over a set of vectors is already a challenging problem. Characterizing the statistical complexity of optimization in this setting (with exact function evaluation) is a surprisingly neglected but fundamental problem. We believe our theoretical results make interesting inroads towards this. The design of algorithms for the noisy query model has to deal with two issues, the optimization problem over the function class and action set, and the complexities of noisy evaluation. We think that making inroads towards the first problem in isolation is a good starting point to understand these problems in a more systematic way.
>
> 6) Minor. Those are great suggestions! We will incorporate them to our writeup.

---

> > ### Author Response · Authors · 2022-11-14
> > **Follow up**
> >
> > Hello,
> >
> > We would like to ping the reviewer to check if they have read our rebuttal and if they have any more questions. We would be happy to reply.
> >
> > The authors

---

> > > ### Comment · Reviewer_UfWC · 2022-11-17
> > > **Response**
> > >
> > > Sorry for the delay!
> > >
> > > > As explained on page 8, figure 1 shows the evolution of the regression...
> > >
> > > I think this is an issue! Can you explain this earlier than page 8?
> > >
> > > > The NN architecture 1500-300 for the autoencoder was chosen via trial and error to ensure the regression loss on VCAP...
> > >
> > > I was looking for something systematic. Given that the architectures are so small, why are you resource constrained? Sorry if I am being dense.
> > >
> > > > Figure labeling we will fix this.
> > >
> > > Thanks.
> > >
> > > > We are not claiming a single method will work over-the-board for all datasets...
> > >
> > > Thanks for the response, and you are right my word choice of "sparse" was wrong.
> > >
> > > > We in fact think of this as a contribution of our work, covering several situations a practitioner may encounter...
> > >
> > > I agree this *should* be a strength but it is not well presented currently (sorry for not mincing words). I think more intuitive diagrams would help.
> > >
> > > > Studying the zero noise setting was an active decision as opposed to a ‘fallback’ option as the reviewer seems to imply.
> > >
> > > I see another reviewer has the same concern I do. I appreciate that there's reasons to look at zero noise and noisy settings, but given that this method is presented as a tool for practitioners, I don't get why the zero noise setting would take precedence. I appreciate that there's a simple translation to a noise-robust version of the algorithm, but I would like to see validation for that prioritized here. Sorry if I am dense here, but isn't work on the zero noise setting only useful if it translates to the noisy setting, which is what we actually encounter in biology?
> > >
> > > > Minor...
> > >
> > > Thanks!

---

> > > > ### Author Response · Authors · 2022-11-17
> > > > **Clarifications - zero noise**
> > > >
> > > > We want to thank the reviewer for their thoughtful feedback. We want to emphasize that just as our title implies the main application we are trying to target in this work is that of genetic perturbation experiments. We believe this is a valid model of this problem setting. Our zero noise assumption is motivated by real-world applications of active learning methods like ours in which selecting the same arm multiple times is almost never done because the noise of that arm is very low relative to the cost of selecting it. In settings like these, arms are almost never selected more than once. In the genetic perturbation context, our reward function is already the average response across technical replicates, each of which is itself an average response across the technical batch of cells (in a single well of a plate) receiving the same genetic perturbation. This average of averages motivates our zero-noise assumption.
> > > >
> > > > A further direction outside the scope of this work would be to explore the noise introduced by biological replicates, for example performing a perturbation from additional human primary cell donors. Neither the CMAP dataset nor the GeneDisco datasets offer the option of observing the same genetic perturbation on a new donor, but as richer genetic perturbation datasets become publicly available across many (10+) donors, we will be excited to extend our approach to accommodate this additional inter-donor source of variability.

---

> > > > > ### Comment · Reviewer_UfWC · 2022-11-17
> > > > > **Response**
> > > > >
> > > > > I see, that makes sense. I will bump your score a touch and keep discussing this with the other reviewers. Could you expand the conclusion with a discussion of this point?

---

> > > > > > ### Author Response · Authors · 2022-11-17
> > > > > > **Thanks so much!**
> > > > > >
> > > > > > Thank you so much for your feedback. We will incorporate the reviewers helpful suggestions to the manuscript.
> > > > > >
> > > > > > The authors.

---

### Official Review · Reviewer_sSRW · 2022-10-24

**Confidence:** 2
**Correctness:** 4
**Technical Novelty And Significance:** 2
**Empirical Novelty And Significance:** 4
**Recommendation:** 8

**Clarity, Quality, Novelty And Reproducibility:**

It is well-written. But I cannot find the reference for citations in 5.0.1 (those datasets)

**Strength And Weaknesses:**

Strong: This is a well-motivated real-world problem. The author gives a meaningful formulation of this problem and shows promising experimental results.

Weaknesses:
1\ The proposed framework in the main paper, although high-level and flexible, seems just a very standard optimism-based framework that incorporates various existing heuristic methods  (Section 4.1, 4.2)

2\ Under the noiseless assumption, the proposed predictor in Eqn.(1) and the corresponding Theorem 4.1 seems not surprising to me -- If there is no misspecification then this is just a standard passive learning problem. If there is some misspecification, then it is linear in misspecification. Also this predictor is not parameter-free because you need to choose tolerance $\gamma$ based on misspecifcation.

**Summary Of The Paper:**

Motivated by the genetic perturbations problem in biology, the authors reformulate this problem as a batched regret minimization problem with a large action space.  Specifically, they assume the feedback is noiseless and prior-known a potential function class that outputs the score of a given action (a perturbation) but this function class can be misspecified.

Based on this setting, they propose the Optimistic Arm Elimination (OAE) framework, which can be adapted to various optimistic predictors. Under this framework, they give a theoretical regret guarantee in terms of the Eluder dimension of the function, the misspecification level, and the batch size. Then they test their framework on various experiments by choosing the regression model as deep neural net and get SOTA in 3 of 4 datasets.

**Summary Of The Review:**

While I believe it is practical and meaningful based on those experimental results. It is a bit unclear to me whether its novelty on theoretical contributions or algorithm design contributions. I am not experienced in doing experiments so it is hard to judge the overall contributions.

Also, given this is an ml for a science paper. I am willing to raise my score if people familiar with this area endorse it.

---

> ### Author Response · Authors · 2022-11-11
> **Response to reviewer sSRW**
>
> We want to thank the reviewer for their kind and detailed comments and would really like to encourage the reviewer to revise their score and engage in a conversation with us the authors during the rebuttal period. We hope the reviewer read the ‘response to all reviewers’ we posted at the top of the page and would love to address any further questions they may have about our work. We are very happy to have a back and forth and will be fast at replying to the reviewers questions if any.
>
> We want to respectfully disagree about the two stated weaknesses the reviewer has mentioned and hope that our discussion can convince the reviewer to change their score.
>
> \textbf{Novelty of our algorithm design contributions}. As the reviewer has identified, the different tractable versions of OAE are based on approximations to the principle of optimism. We want to reiterate as we also did in the paper that we do not consider our main contribution is not the use of optimism, but the development and analysis of tractable ways of doing so with neural networks (NN).  We don’t see it as a limitation to bring to life in the realm of neural network approximation a universally known principle such as optimism. If anything, we consider this to be one of our main contributions: to devise several novel ways of implementing optimistic data collection strategies when using neural network models. We are particularly proud of developing the optimistic regularization method. Optimistic regularization is arguably more in tune with theoretical results than other approximate optimistic methods such as ensembles. This is because it is directly based on finding via \textbf{differentiable optimization} a function whose fit is ‘good’ in the data we have collected so far, while its responses are optimistic on the unseen data points. We are not aware of other work that uses a similar approach that can be easily scaled to very large dimensional data and large NN architectures. Moreover, in contrast with ensemble methods, optimistic regularization only requires a single model. Having multiple large NN models in memory could be unbearably expensive due to memory constraints in many applications. We are also very proud of the sequential batch optimism methods that we have introduced for batch data collection, and how our NN methods easily extend to this setting. In summary, we believe that (in particular) the simple optimistic regularization technique we have introduced can be of great benefit to the community, not only in the setting of genetic perturbation experiments but beyond.
>
>
> \textbf{Theoretical results}. Here we want to respectfully push back against the reviewers claims. Our theoretical results are highly non-trivial. First, recall that in the stated theorem (Theorem 4.1) the regret upper bound scales as $\mathcal{O}\left( d b + \omega \sqrt{d}
> T b \right)$. We want to draw attention to the square root dependence on $d$  in the linear misspecification component of the regret. This is very significant because it generalizes the results of [1] to the setting of function approximation (beyond linear) and when the number of queries per round equals $b$, possibly more than $1$. The main ingredient in our proof (Proposition D.4) is a completely novel result. It is \emph{not} a simple rehashing of Proposition 4 in [2] but a unique result on its own. We encourage the reviewer to compare it with the results in [2] to see how the proof results and techniques are different. In particular, the proof of Proposition D.4 involves a careful and novel use of Hall’s marriage theorem in this setting. Moreover, Proposition D.4 can be used to show the regret in noisy settings and for example when $\omega = 0$, scales as $\mathcal{O}\left( d \sqrt{Tb} + bd \right)$ this again is significant because a naive analysis of a parallel non linear bandit procedure would scale as $\mathcal{O}\left( d b \sqrt{T}\right)$ (linearly with $b$). Instead, with Proposition D.4 we can show the parallel regret scales as the sequential regret of running Tb sequential queries (i.e. the b dependence is of order $\sqrt{b}$) up to a burn in factor of order db. This can be done by simply substituting our novel result (Proposition D.4) into the proof of Lemma 2 in [2]. In other words, the cost of parallelism only emerges at the beginning. This generalizes for example all the results of [3] since it would yield results for parallel exploration (b  > 1) with nonlinear functions.  \textit{Parameter free} This is indeed true, and it is a major issue in \emph{many} bandit algorithms. In this case this can be mitigated by simply using an online model selection wrapper around our algorithm, which for selecting the misspecification parameter will achieve near optimal oracle rates.
>
> [1] Learning with Good Feature Representations in Bandits and in RL with a Generative Model.
>
> [2] Eluder Dimension and the Sample Complexity of Optimistic Exploration.
>
> [3] Parallelizing Contexual Linear Bandits.

---

> > ### Author Response · Authors · 2022-11-14
> > **Follow up**
> >
> > Hello,
> >
> > We would like to ping the reviewer to check if they have read our rebuttal and if they have any more questions. We would be happy to reply.
> >
> > The authors

---

> > > ### Author Response · Authors · 2022-11-17
> > > **Follow up**
> > >
> > > Dear Reviewer,
> > >
> > > We would like to follow up to inquire if the reviewer has any questions. We hope our explanation above satisfied the reviewer. We would also like to note that other reviewers have revised their scores upward. We hope our response can convince this reviewer to do the same.
> > >
> > > Thanks so much,
> > >
> > > The Authors

---

> > > > ### Author Response · Authors · 2022-11-18
> > > > **an additional note**
> > > >
> > > > We want to thank the reviewer for their thoughtful feedback. We want to emphasize that just as our title implies the main application we are trying to target in this work is that of genetic perturbation experiments. We believe this is a valid model of this problem setting. Our zero noise assumption is motivated by real-world applications of active learning methods like ours in which selecting the same arm multiple times is almost never done because the noise of that arm is very low relative to the cost of selecting it. In settings like these, arms are almost never selected more than once. In the genetic perturbation context, our reward function is already the average response across technical replicates, each of which is itself an average response across the technical batch of cells (in a single well of a plate) receiving the same genetic perturbation. This average of averages motivates our zero-noise assumption.
> > > >
> > > > A further direction outside the scope of this work would be to explore the noise introduced by biological replicates, for example performing a perturbation from additional human primary cell donors. Neither the CMAP dataset nor the GeneDisco datasets offer the option of observing the same genetic perturbation on a new donor, but as richer genetic perturbation datasets become publicly available across many (10+) donors, we will be excited to extend our approach to accommodate this additional inter-donor source of variability.

---

> > > > > ### Author Response · Authors · 2022-11-23
> > > > > **Follow up**
> > > > >
> > > > > Dear Reviewer,
> > > > >
> > > > > We would like to follow up with our comments and ask if there are any further questions. Would you be so kind, if our responses have satisfied you, to increase your score?
> > > > >
> > > > > The Authors

---

> > > > > ### Comment · Reviewer_sSRW · 2022-11-27
> > > > > **Feedback**
> > > > >
> > > > > Sorry for the late reply.
> > > > >
> > > > > Thanks for the clarification on your theoretical result in Theorem 4.1. I agree it is a nontrivial contribution.
> > > > >
> > > > > I am not sure how this optimistic regularization can help other areas besides the current setting, since there is still a gap between theory and practice. It might be better to empirically compare it with some other online active learning algorithms which combine uncertainty measurement as well as regret minimization. Nevertheless, I agree that this can be a future focus and the current methods always give enough technical contribution.
> > > > >
> > > > > Overall, I think it is good paper but my evaluation is purely from the theoretical perspective.

---

### Decision · Program_Chairs · 2023-01-20

**Decision:**

Accept: notable-top-25%

**Justification For Why Not Higher Score:**

- Concerns about the appropriateness of a zero-noise assumption, both in the application domain considered, and in other applications

**Justification For Why Not Lower Score:**

- Very interesting problem & experimental evaluation
- High-quality analysis with a well-justified method

**Metareview: Summary, Strengths And Weaknesses:**

The papers considers designing genetic modifications to cells to engineer their phenotype.  Evaluating genetic modifications is time-consuming and expensive. The paper develops a iterative batch method for exploring the space of modifications.  This method is based on the proposed Optimistic Arm Elimination principle. The paper presents an analysis of its convergence in terms of the Eluder dimension of the function mapping modifications to phenotypes.  The method is evaluated on an array of interesting datasets.

Strengths
- The application domain is important and interesting
- The paper develops a scalable and tractable way to use the principle of optimism (common in bandits) within the context of neural network approximations
- The proposed method is accompanied by a non-trivial regret guarantee
- The evaluation is performed on a variety of interesting datasets
- The paper is clearly written overall

Weaknesses
- Analysis assumes noiseless responses
- Some minor opportunities to improve clarity and clarify experiments

**Note From Pc:**

if the above contains the word "oral" or "spotlight" please see: "oral" presentation means -> notable-top-5% and "spotlight" means -> notable-top-25%. As stated in our emails, we are disassociating presentation type from AC recommendations